# SNORD90 induces glutamatergic signaling following treatment with monoaminergic antidepressants

Rixing Lin[1,2†], Aron Kos[3,4,5†], Juan Pablo Lopez[3,4,5], Julien Dine[3,4,5], Laura M Fiori[1], Jennie Yang[1], Yair Ben-Efraim[3,4,5], Zahia Aouabed[1], Pascal Ibrahim[1,2], Haruka Mitsuhashi[1,2], Tak Pan Wong[6,7], El Cherif Ibrahim[8], Catherine Belzung[9], Pierre Blier[10], Faranak Farzan[11], Benicio N Frey[12,13], Raymond W Lam[14], Roumen Milev[15], Daniel J Muller[16,17], Sagar V Parikh[18], Claudio Soares[15], Rudolf Uher[19,20], Corina Nagy[1], Naguib Mechawar[1], Jane A Foster[12,13,16], Sidney H Kennedy[16,21], Alon Chen[3,4,5], Gustavo Turecki[1]*

[1]McGill Group for Suicide Studies, Douglas Mental Health University Institute, Department of Psychiatry, McGill University, Montreal, Canada; [2]Integrated Program in Neuroscience, McGill University, Montreal, Canada; [3]Department of Stress Neurobiology and Neurogenetics, Max Planck Institute of Psychiatry, Munich, Germany; [4]Department of Brain Sciences, Weizmann Institute of Science, Rehovot, Israel; [5]Department of Molecular Neuroscience, Weizmann Institute of Science, Rehovot, Israel; [6]Neuroscience Division, Douglas Research Centre, Montreal, Canada; [7]Department of Psychiatry, McGill University, Montreal, Canada; [8]Aix-Marseille Université, CNRS, INT, Institute Neuroscience Timone, Marseille, France; [9]UMR 1253, iBrain, UFR Sciences et Techniques; Parc Grandmont, Tours, France; [10]Mood Disorders Research Unit, University of Ottawa Institute of Mental Health Research, Ontario, Canada; [11]eBrain Lab, Simon Fraser University, Columbia, Canada; [12]Department of Psychiatry and Behavioural Neurosciences, McMaster University, Hamilton, Canada; [13]Mood Disorders Program, St. Joseph's Healthcare Hamilton, Hamilton, Canada; [14]Department of Psychiatry, University of British Columbia, Columbia, Canada; [15]Departments of Psychiatry and Psychology, Queens University, Ontario, Canada; [16]Department of Psychiatry, University Health Network, Krembil Research Institute, University of Toronto, Toronto, Canada; [17]Centre for Addiction and Mental Health, Toronto, Canada; [18]Department of Psychiatry, University of Michigan, Ann Arbor, United States; [19]Nova Scotia Health Authority, Halifax, Canada; [20]Department of Psychiatry, Dalhousie University, Halifax, Canada; [21]St Michael's Hospital, Li Ka Shing Knowledge Institute, Centre for Depression and Suicide Studies, Toronto, Canada

*For correspondence:
gustavo.turecki@mcgill.ca

†These authors contributed equally to this work

**Abstract** Pharmacotherapies for the treatment of major depressive disorder were serendipitously discovered almost seven decades ago. From this discovery, scientists pinpointed the monoaminergic system as the primary target associated with symptom alleviation. As a result, most antidepressants have been engineered to act on the monoaminergic system more selectively, primarily on serotonin, in an effort to increase treatment response and reduce unfavorable side effects. However, slow and inconsistent clinical responses continue to be observed with these available treatments. Recent findings point to the glutamatergic system as a target for rapid acting antidepressants. Investigating different cohorts of depressed individuals treated with serotonergic

and other monoaminergic antidepressants, we found that the expression of a small nucleolar RNA, *SNORD90*, was elevated following treatment response. When we increased *Snord90* levels in the mouse anterior cingulate cortex (ACC), a brain region regulating mood responses, we observed antidepressive-like behaviors. We identified neuregulin 3 (*NRG3*) as one of the targets of *SNORD90*, which we show is regulated through the accumulation of N$^6$-methyladenosine modifications leading to YTHDF2-mediated RNA decay. We further demonstrate that a decrease in NRG3 expression resulted in increased glutamatergic release in the mouse ACC. These findings support a molecular link between monoaminergic antidepressant treatment and glutamatergic neurotransmission.

## Editor's evaluation

This is an important study that uncovers a new molecular pathway that links traditional monoaminergic antidepressants with regulation of glutamate neurotransmission. The data provided for the model are convincing and demonstrate the pathway in human plasma and brain, mouse brain, and cultured cells, using the relative strengths of each system. The work will be of interest to psychiatrists studying depression as well as basic neurobiologists interested in monoamine signaling in the brain.

## Introduction

Antidepressants are the first-line treatment for major depressive disorder (MDD), collectively accounting for one of the most prescribed medications (*Brody and Gu, 2020*). However, response to antidepressant treatment is variable with less than 50% of patients responding to first trial, and up to 40% with no clinical response after two or more trials (*Cipriani et al., 2018*). While most currently available antidepressants act by selectively or non-selectively targeting serotonergic receptors the exact mechanisms by which they effect mood changes and improvements in depression remain unknown. Importantly, although the enhancement of monoamine function, in particular serotonin, can be observed within hours after antidepressant drug administration, clinical improvements are not observed until days or weeks following antidepressant treatment initiation (*Harmer et al., 2017*; *Ross and Renyi, 1969*; *Vetulani and Sulser, 1975*). The delayed clinical response led researchers to study underlying neurobiological adaptations to understand mechanisms of antidepressant response. Monoamines have been a focus of depression studies since most treatment options target this system. However, given the delayed clinical response other neurotransmitter systems have been gaining interest in depression; particularly the glutamatergic system, which is believed to be the target of rapid acting antidepressants (*Berman et al., 2000*; *Duman, 2018*). Moreover, recent evidence suggest that monoaminergic antidepressants may act by modulating the glutamatergic system, although it is still unclear through what molecular mechanisms (*Bonanno et al., 2005*; *Musazzi et al., 2013*).

The functional activity of genes is at the core of all biological processes. Thus, investigating the molecular factors that modulate gene expression in relation to antidepressant treatment should provide better insight into the molecular mechanisms surrounding antidepressant response. Non-coding RNAs act as fine tuners of gene expression through a diverse range of functional mechanisms (*Statello et al., 2021*). In this study, we focused our attention on a class of small non-coding RNA called small nucleolar RNAs (snoRNAs). Although snoRNAs have classically been associated with housekeeping roles they have more recently been shown to be involved in complex regulatory roles in gene expression such as regulation of alternative splicing, precursor to smaller miRNA-like RNA fragments, and direct regulation of mRNA expression (*Kishore and Stamm, 2006*; *Ender et al., 2008*; *Sharma et al., 2016*). Here, we identified a molecular mechanism whereby monoaminergic antidepressants produce an effect on glutamatergic neurotransmission. We found elevated levels of a snoRNA, *SNORD90*, in response to antidepressant drug exposure and report that *SNORD90* guides N$^6$-methyladenosine (m6A) modifications onto neuregulin 3 (NRG3), which in turn lead to YTHDF2-mediated down-regulation of NRG3 expression and subsequent increases in glutamatergic release.

## Results

### *SNORD90* levels are increased in response to monoaminergic antidepressants

We initially examined snoRNA expression using small RNA-sequencing data from peripheral blood samples collected from three independent antidepressant clinical trials (discovery cohort, replication cohort 1 and replication cohort 2) administrating duloxetine, escitalopram, or desvenlafaxine. To identify snoRNAs associated with treatment response, we focused our attention on snoRNAs that were differentially expressed between baseline (T0) and eight weeks after treatment (T8), when clinical outcome was ascertained. In doing so, we identified nine snoRNAs in our placebo controlled double blind discovery cohort that had a significant interaction (p<0.05) between time (T0/T8) and treatment outcome (response/non-response; *Figure 1A*, *Supplementary file 1*). In particular, *SNORD90* was the only snoRNA that was consistently up-regulated after antidepressant treatment in subjects who responded to treatment across all three independent cohorts (*Figure 1B*, *Supplementary files 1–3* & *Figure 1—figure supplement 1A*). As such, we further investigated *SNORD90*.

To better understand if *SNORD90* expression has similar changes in the brain following antidepressant treatment as observed in peripheral tissue, we investigated human post-mortem anterior cingulate cortex (ACC), a brain region that plays an important role in the regulation of mood (*Mayberg et al., 2005*; *Roet et al., 2020*). We studied individuals who died while affected with MDD and were or were not treated with antidepressants. We observed a specific up-regulation of *SNORD90* in individuals who were depressed when they died and were actively treated with antidepressants (*Figure 1C*). To further asses if there is any cell type specificity to the effects of antidepressants on the up-regulation of *SNORD90*, we employed fluorescence-activated nuclei sorting (FANS) using samples from the human post-mortem ACC tissue described above. We separated neuronal (NeuN +nuclei) and non-neuronal (NeuN- nuclei), and measured the expression of *SNORD90* in each nuclei fraction. In both the neuronal and non-neuronal nuclei fractions we observed the same pattern of expression as in bulk tissue, where *SNORD90* was up-regulated in individuals with depression that were actively treated with antidepressants at the time of death (*Figure 1—figure supplement 1B*). These results suggest that antidepressants do not up-regulate *SNORD90* in a cell-type specific manner. To follow up these results, we investigated the expression of *Snord90* in an unpredicted chronic mild stress (UCMS) mouse model, which is commonly used to study depressive-like behaviors in mice. Male mice were subjected to UCMS, followed by administration of fluoxetine (*Hervé et al., 2017*). We specifically profiled the mouse cingulate area 1/2 (cg1/2), a region that is equivalent to the human ACC. Similar to the results observed in humans, we observed a specific up-regulation of *Snord90* in the ACC of mice that underwent the UCMS paradigm followed by antidepressant administration, whereas UCMS or antidepressant administration alone did not alter the expression of *Snord90* (*Figure 1D*). To assess if the effects observed on the expression of *SNORD90* were specific to antidepressants or common to other drugs, we treated human neuronal cells derived from iPSCs and differentiated to a monoaminergic phenotype with several psychotropic drugs, including duloxetine, escitalopram, haloperidol, lithium, or non-psychotropic drugs (aspirin) and regular culture media. We observed that *SNORD90* expression was up-regulated exclusively by antidepressant drugs, while other treatments did not significantly alter *SNORD90* expression (*Figure 1E*). Together our data suggests that antidepressant treatment response associates with an increase in *SNORD90* expression.

### *Snord90* over-expression in mouse ACC induces anti-depressive like behaviours

Given that we observed an upregulation of *SNORD90* with antidepressant treatment response in humans and mice, we next investigated if the over-expression of *Snord90* in the cg 1/2 cortex of mice has behavioural implications. We over-expressed *Snord90* or a full scrambled *Snord90* sequence control (*Supplementary file 4*) in male mouse cg1/2 via bilateral injections of an adeno-associated virus (AAV) followed by a battery of behavioural tests designed to measure anxiety and depressive-like behaviors in mice (*Figure 2A*). We did not observe any differences in the total distance traveled in the open field test, which indicated no general locomotor differences between *Snord90* and scramble over-expression (*Figure 2B*). *Snord90* over-expression increased the amount of time spent in the open arm and decreased the amount of time spent in the closed arm of the elevated plus maze (*Figure 2C*).

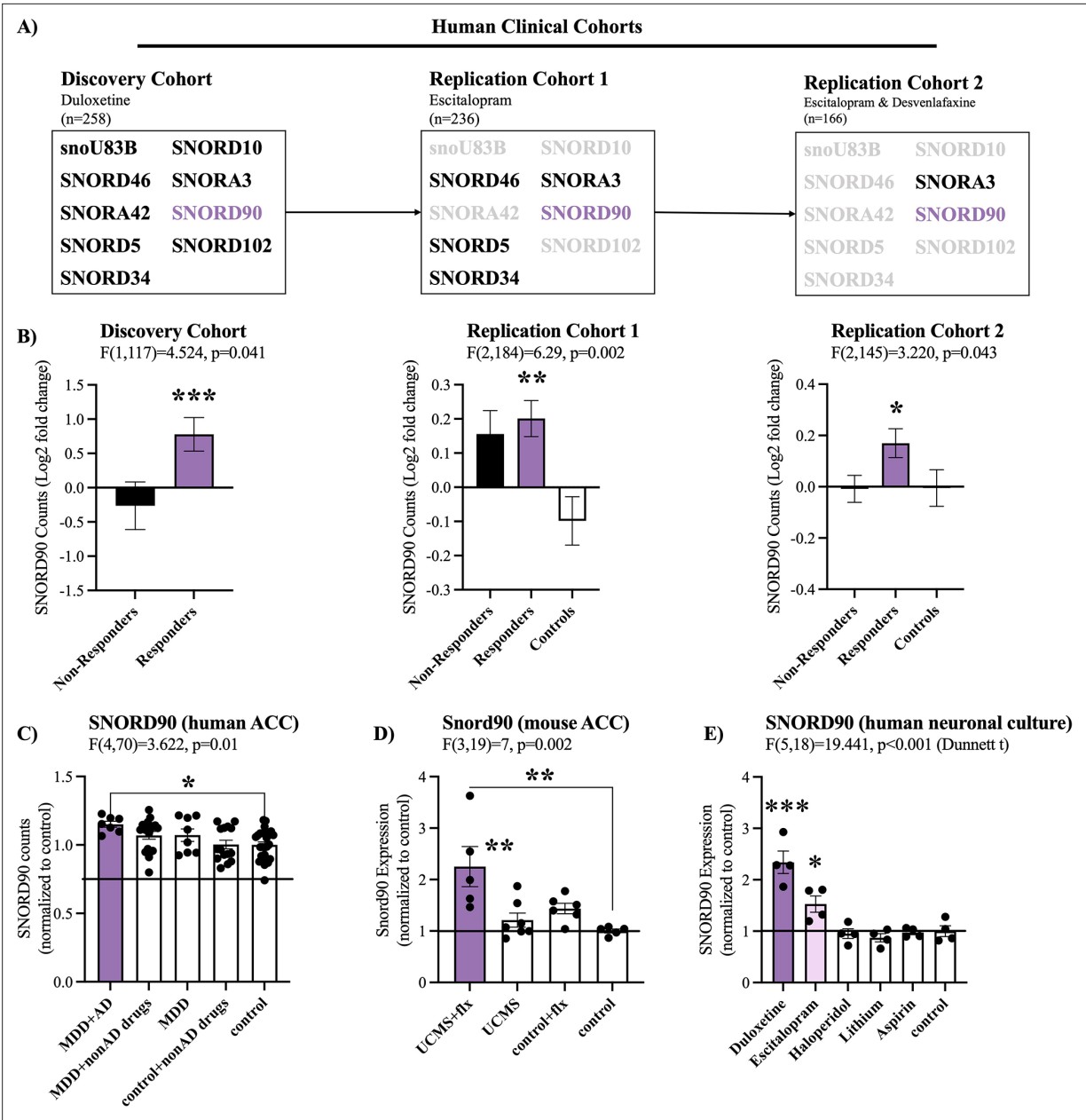

**Figure 1.** *SNORD90* expression is associated with antidepressant treatment response. (**A**) Two-way mixed multivariable ANOVA indicates a significant interaction between clinical response (responders/non-responders; between-factor) and treatment course (T0/T8; within factor). Nine snoRNAs displayed significant effects in the discovery cohort. Five out of the nine snoRNAs were replicated in replication cohort 1 with *SNORD90* and *SCARNA3* further replicated in replication cohort 2. (**B**) Log2 fold-change of the expression of *SNORD90* before and after antidepressant treatment for all three clinical cohorts. *SNORD90* displayed significantly increased expression after eight weeks of antidepressant treatment specifically in those who responded across all three clinical cohorts. See *Figure 1—figure supplement 1A* for *SCARNA3* expression (**C**) *SNORD90* expression in human post-mortem ACC. Using toxicology screens, each sample was separated into the following groups: MDD with presence of antidepressants (MDD +AD; n=7), MDD with presence of non-antidepressant drugs (MDD +nonAD drugs; n=18), MDD with negative toxicology screen (MDD; n=8), controls with non-antidepressant drugs (control +nonAD drugs; n=15), and controls with negative toxicology screens (control; n=26). No control samples were positive for antidepressant drugs. (**D**) *Snord90* expression in the ACC of mice that underwent unpredictable chronic mild stress (UCMS) and fluoxetine (flx) administration (UCMS + flx, n=5; UCMS, n=7; control +flx, n=6; control, n=5). (**E**) *SNORD90* expression in human neuronal cultures exposed to various psychotropic drugs (n=4 per group). All bar plots represent the mean with individual data points as dots. Error bars represent S.E.M. One-way ANOVA with Bonferroni post-hoc (unless otherwise indicated on the graph). *p<0.05, **p<0.01, ***p<0.001.

The online version of this article includes the following figure supplement(s) for figure 1:

*Figure 1 continued on next page*

Figure 1 continued

**Figure supplement 1.** *SCARNA3* expression in human clincial cohorts and *SNORD90* expression in neuronal and non-neuronal cell types from human post-mortem ACC.

**Figure supplement 2.** *NRG3* expression is negatively associated with *SNORD90* expression.

Moreover, *Snord90* over-expression increased time spent grooming after splashing with 10% sucrose solution (*Figure 2D*) and increased time spent struggling in the tail suspension test (*Figure 2E*). Lastly, we calculated an integrated emotionality Z-score by combining data from the elevated plus maze, splash test, and tail suspension test (*Figure 2F*). Overall, our results consistently showed that over-expressing *Snord90* levels in cg1/2 yielded decreased emotionality (*Figure 2F*) indicative of a decrease in anxiety-like and depressive-like behaviours.

## SNORD90 directly down-regulates NRG3

*SNORD90* is subcategorized as an orphan snoRNA as it does not have any canonical rRNA, snRNA, or tRNA targets (*Lestrade, 2006*). However, more recent studies have indicated that some snoRNAs exhibit atypical functioning such as regulation of alternative splicing and modulating expression of mRNA (*Kishore and Stamm, 2006*; *Ender et al., 2008*; *Sharma et al., 2016*). Thus, we explored possible RNA targets for *SNORD90* using the basic local alignment (BLAST) search tool for base complementarity (*Supplementary file 5*), and using the C/D box snoRNA target prediction tool 'PLEXY' (*Supplementary files 6-7*; *Kehr et al., 2011*). Using these in silico methods, we identified several predicted targets for *SNORD90* and selected neuregulin 3 (NRG3) as a putative gene target because it was the only candidate gene to be predicted by both in silico approaches (*Supplementary files 5-6*; see Materials and methods for more details). More specifically, we identified three predicted binding regions for *SNORD90* on NRG3 pre-mRNA (*pre-NRG3*) (*Figure 3A*). NRG3 is a growth factor that is part of the neuregulin family which is highly enriched in the brain and has been previously associated with psychiatric phenotypes (*Paterson et al., 2017*; *Wang et al., 2018*; *Zhang et al., 1997*). To determine if *SNORD90* regulates *NRG3*, we first assessed *NRG3* expression in the same human and mouse ACC samples, as well as in the neuronal cultures described above (*Figure 1—figure supplement 2A–C*). We observed a negative correlation between *SNORD90* and *NRG3* in all three experimental contexts (*Figure 1—figure supplement 2D–F*), and interestingly, we observed the most significant differences in direction of fold-change between *SNORD90* and *NRG3* expression in groups with antidepressant drug exposure (*Figure 1—figure supplement 2G–I*).

To further investigate the strength of our in silico target prediction approach, we directly manipulated *SNORD90* expression levels in human neural progenitor cells (NPCs) and assessed the resulting effects on *NRG3*, the only gene to be predicted by both in silico approaches, as well as the top predicted targets by each of our in silico approaches: ST6 N-acetylgalactosaminide alpha-2,6-sialyltransferase 3 (*ST6GALNAC3*), Rho GTPase activating protein 29 (*ARHGAP29*), low-density lipoprotein receptor class A domain containing 4 (*LDLRAD4*), endoglin (*ENG*), exocyst complex component 6B (*EXOC6B*), and SPARC (osteonectin), cwcv and kazal like domains proteoglycan 3 (*SPOCK3*) (*Supplementary files 5-6*). We constructed AAV vectors over-expressing wild-type *SNORD90*, as well as two scrambled controls and transfected into human NPCs in culture (*Figure 3—figure supplement 1A–C* & *Supplementary file 4*). The first control had a scramble sequence in the central region of the *SNORD90* transcript, where the predicted complementary sequence to its predicted targets lies, and the second control had a full scramble of the entire *SNORD90* transcript (*Figure 3—figure supplement 1A* & *Supplementary file 4*). Over-expressing *SNORD90* resulted in a~50% decrease in NRG3 expression at both the mRNA (*Figure 3B*) and protein levels (*Figure 3—figure supplement 1D*). Seed scramble and full scramble over-expression did not alter the expression of *NRG3*, indicating the region of predicted complementarity between *SNORD90* and *NRG3* plays an important role in the ability of *SNORD90* to down-regulate *NRG3* (*Figure 3B*, *Figure 3—figure supplement 1*). Furthermore, *SNORD90* over-expression also resulted in a~70% increase in *ENG* expression (*Figure 3—figure supplement 1E*), while the other five predicted targets did not show significant differences in expression (*Figure 3—figure supplement 1F–J*). We next examined the effects of *SNORD90* knock-down using antisense oligonucleotides (ASO). We screened four ASOs that target different regions of *SNORD90* as well as an ASO scramble control (*Figure 3—figure supplement 1K* & *Supplementary file 8*) and selected the ASO achieving the best knock-down of *SNORD90* (*Figure 3—figure supplement 1L*). *SNORD90*

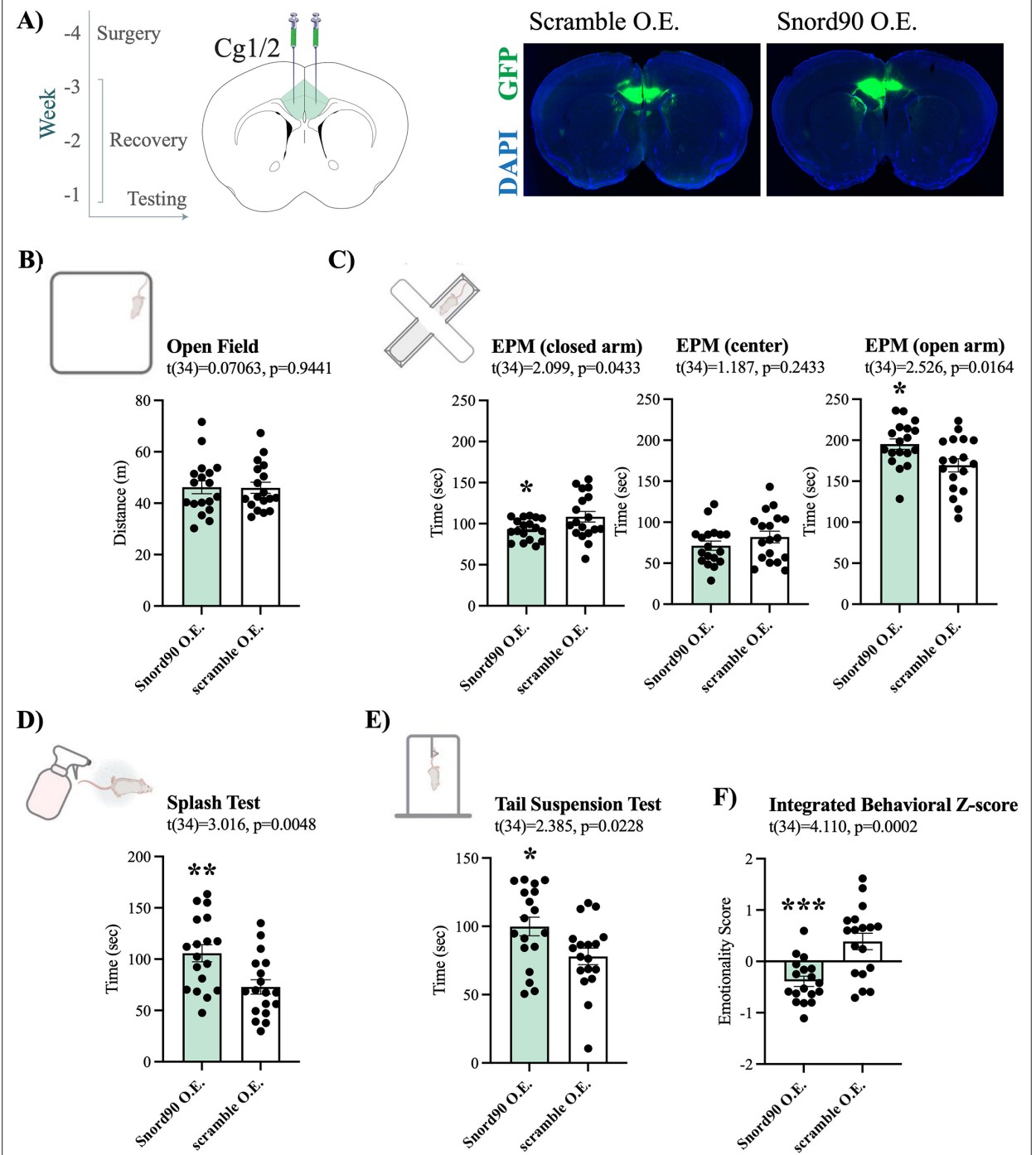

**Figure 2.** *Snord90* over-expression in mouse Cg1/2 induces anxiolytic and anti depressive-like behaviors. (**A**) Timeline of experimental procedure with time (weeks) in relation to behavioral testing. Surgery for viral injection was performed followed by 3 weeks of recovery before behavioral testing (left). Coronal diagram of the mouse brain representing viral injection of *Snord90* expression vector (Snord90 O.E., n=18) or scramble control expression vector (scramble O.E., n=18) into Cg1/2 (mouse equivalent to human ACC) (center). Representative images of GFP expression indicating site specific expression of each construct (right). (**B**) The open field test showing total distance traveled in meters. (**C**) The elevated plus maze test with total time spent in the closed arms, center, and open arms of the plus maze. (**D**) The splash test (10% sucrose solution) with total grooming time. (**E**) The tail suspension test with total struggling time. (**F**) Emotionality z-score integrating the elevated plus maze, splash test, and tail suspension test. All bar plots represent the mean with individual animals as dots. Error bars represent S.E.M. Student's two-tailed t-test. *p<0.05, **p<0.01.

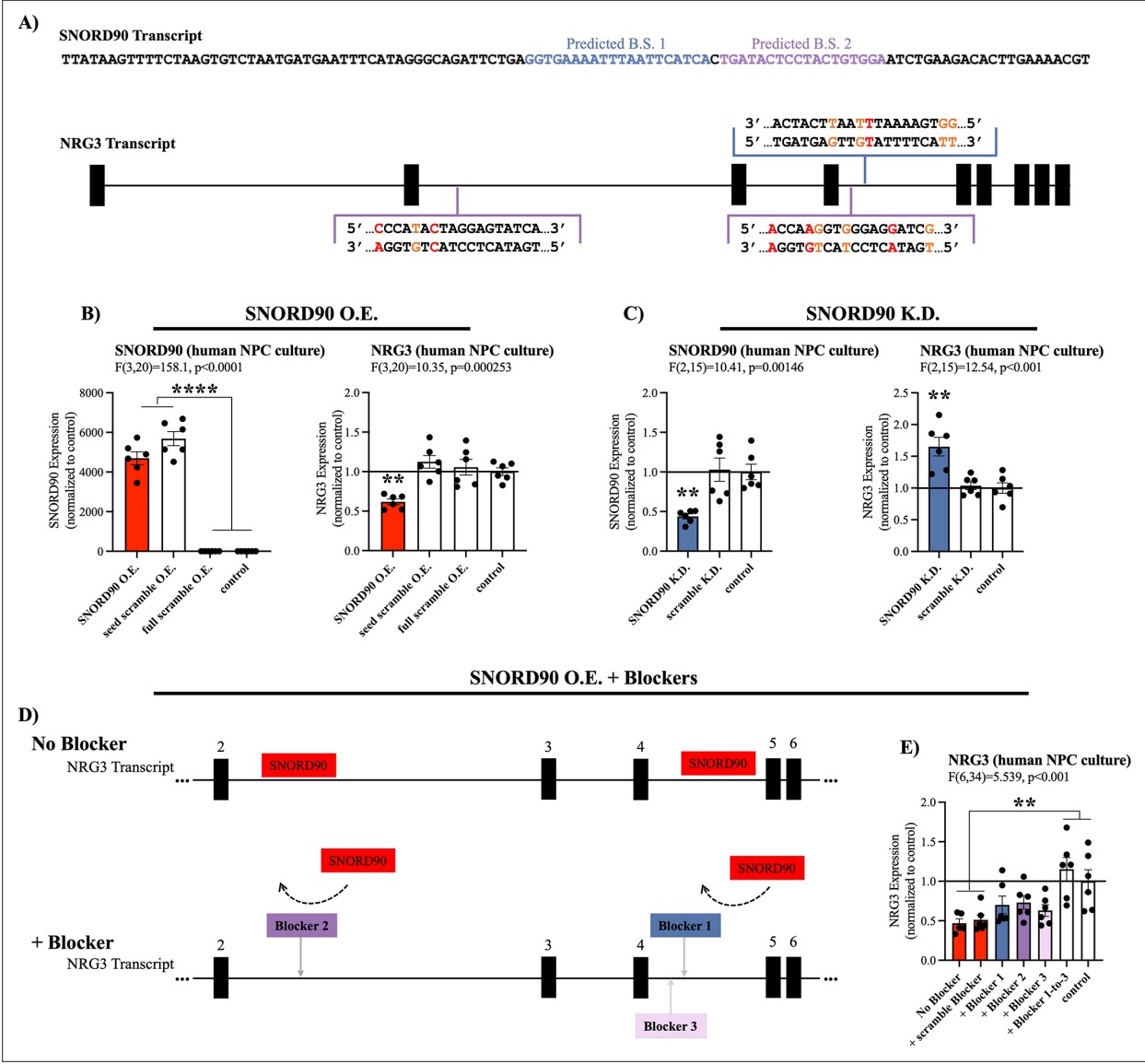

**Figure 3.** *SNORD90* down-regulates *NRG3*. (**A**) Full sequence of mature *SNORD90* transcript with highlighted regions labeled predicted binding site (B.S.) 1 and predicted B.S. 2, which are predicted to bind to *NRG3*. Schematic representation of NRG3 pre-mRNA transcript indicating regions on *NRG3* where *SNORD90* is predicted to bind. The color of the bracket corresponds to predicted B.S.-1 or predicted B.S.-2. (red nucleotide indicates mismatch, orange nucleotide indicates G-T wabble pair). (**B**) Expression of *SNORD90* (left) and *NRG3* (right) in human NPCs after transfection with *SNORD90* expression vector (SNORD90 O.E., n=6), seed scramble expression vector (seed scramble O.E., n=6), and full scramble expression vector (full scramble O.E., n=6). Non-transfected NPCs under normal culture conditions as control (n=6) (**C**) Expression of *SNORD90* (left) and *NRG3* (right) after transfecting human NPCs with an antisense oligonucleotides (ASO) designed to knock-down *SNORD90* (SNORD90 K.D., n=6) or scrambled ASO (scramble K.D., n=6). Non-transfected NPCs under normal culture conditions as control (n=6). (**D**) Schematic representation of co-transfection of *SNORD90* over-expression vector without target blockers (top) and with target blockers (bottom). Target blockers designed to bind to regions on *pre-NRG3* where *SNORD90* is predicted to bind, consequently blocking *SNORD90* from interacting with those regions on *NRG3* (bottom). (**E**) *NRG3* expression following *SNORD90* expression vector co-transfected with a scrambled blocker (+scramble blocker, n=6), target blockers one site at a time (+Blocker 1, n=6;+Blocker 2, n=6, and +Blocker 3, n=6), or all three blockers simultaneously (+Blocker1-to-3, n=6). Non-transfected NPCs under normal culture conditions as control (n=6). All bar plots represent the mean with individual data points as dots. Error bars represent S.E.M. One-way ANOVA with Bonferroni post-hoc. **p<0.01, ****p<0.0001.

The online version of this article includes the following source data and figure supplement(s) for figure 3:

**Figure supplement 1.** *SNORD90* over-expression and knock-down in human NPC culture.

**Figure supplement 1—source data 1.** Original uncropped western blots for NRG3 and GAPDH found in *Figure 3—figure supplement 1D*.

knock-down by approximately 55% resulted in a~50% up-regulation of *NRG3* expression (*Figure 3C*). In addition, SNORD90 knock-down also resulted in a~20% reduction of *ENG* expression (*Figure 3—figure supplement 1M*), while the other five predicted targets again did not show consistent significant differences in expression (*Figure 3—figure supplement 1N–R*). This was particularly interesting since our results indicate that *SNORD90* can both up-regulate and down-regulate target transcripts. These results also suggest that PLEXY has better prediction accuracy since both transcripts displaying robust significant expression changes were predicted by the PLEXY algorithm (*Supplementary file 6*). ENG is a type I transmembrane glycoprotein, part of the transforming growth factor beta (TGF-β) receptor complex and is primarily expressed with activated endothelial cells playing a major role in angiogenesis both in development and tumor progression (*Schoonderwoerd et al., 2020*). Since NRG3 is primarily expressed in the central nervous system and has been linked to psychiatric disorders in previous studies, we focused our attention exclusively on NRG3 in downstream experiments.

Given that our in-silico predictions indicate that *SNORD90* is primarily interacting with intronic regions of *pre-NRG3*, we next investigated that expression of *pre-NRG3* in the same experiments detailed above; however, *pre-NRG3* expression was not altered by *SNORD90* over-expression or knock-down (*Figure 3—figure supplement 1S–T*). Finally, to further confirm the importance of direct interaction between *SNORD90* and *pre-NRG3*, we designed target blockers which have sequence complementarity to the predicted *SNORD90* binding sites on *pre-NRG3* (*Figure 3D*, *Supplementary file 9*). Target blockers were co-transfected with our *SNORD90* expression vectors (*Figure 3D*). Interestingly, when each individual site was blocked, we observed a partial rescue of *NRG3* expression (*Figure 3E*). However, simultaneously blocking all three predicted sites was required for a complete rescue of the down-regulation effects of over-expressing *SNORD90* (*Figure 3E*). *Pre-NRG3* expression was not altered by any target blockers (*Figure 3—figure supplement 1U*). Together our data suggests that *SNORD90* directly down-regulates NRG3 expression through multiple interaction sites on intronic regions of *pre-NRG3*.

## *SNORD90* associates with RBM15B and guides m6A modifications onto *NRG3*

We next asked what mechanisms may explain *SNORD90's* effects on *NRG3* given that it interacts with intronic regions of *pre-NRG3*, and yet, it does not affect *pre-NRG3* expression. Since *SNORD90* displays atypical functioning, we posited that it could regulate *NRG3* through the recruitment of a unique set of RNA binding proteins (RBPs) that is different from canonically functioning C/D box snoRNAs (*Falaleeva et al., 2016*). Using the in silico tool 'oRNAment', we identified a sequence-motif for RNA Binding Motif Protein 15B (RBM15B); this motif sequence was further confirmed by *Van Nostrand et al., 2020* (*Supplementary file 10*; *Benoit Bouvrette et al., 2020*; *Van Nostrand et al., 2020*). RBM15B is a key regulator of RNA N$^6$-methyladenosine (m6A) modifications by facilitating interactions with Wilms' tumor 1-associating protein (WTAP), which in turn binds to methyltransferase like 3 (METTL3) forming a major m6A writer complex (*Patil et al., 2016*). To confirm the association of *SNORD90* with RBM15B, we performed RNA immunoprecipitation (RIP) against RBM15B as well as the canonical core enzymatic protein for C/D snoRNAs, Fibrillarin (*Figure 4A*; *Tollervey et al., 1991*; *Tollervey et al., 1993*). We observed a significantly higher level of association of *SNORD90* to RBM15B compared to fibrillarin, whereas a canonically functioning snoRNA, *SNORD44*, displayed enrichment for fibrillarin IP (*Figure 4A*). Furthermore, when we over-expressed *SNORD90* we observed an increase in association of *SNORD90* with RBM15B, compared to over-expression of a scrambled control (*Figure 4B*). We did not observe an increase of *SNORD90* association with fibrillarin following *SNORD90* over-expression, indicating that *SNORD90* showed preference for interacting with RBM15B (*Figure 4B*). Since snoRNAs and RBM15B are both found in the nucleus this may, in part, explain *SNORD90's* role in interacting with intronic regions of *pre-NRG3* (*Falaleeva et al., 2016*; *Patil et al., 2016*). Canonical snoRNAs function by guiding their partner proteins to target transcripts which, in-turn, induce a chemical modification (*Kufel and Grzechnik, 2019*). Thus, we hypothesized that *SNORD90* is likely acting as a guide RNA for RBM15B and its associated m6A writer complex, resulting in an increase in m6A levels on *NRG3*. To test this hypothesis, we measured total m6A abundance on *pre-NRG3* and *NRG3* from our in vitro human NPC culture experiments over-expressing *SNORD90* detailed above. We observed an increase of m6A abundance on both *pre-NRG3* and *NRG3*, following *SNORD90* over-expression, whereas seed scramble and full scramble

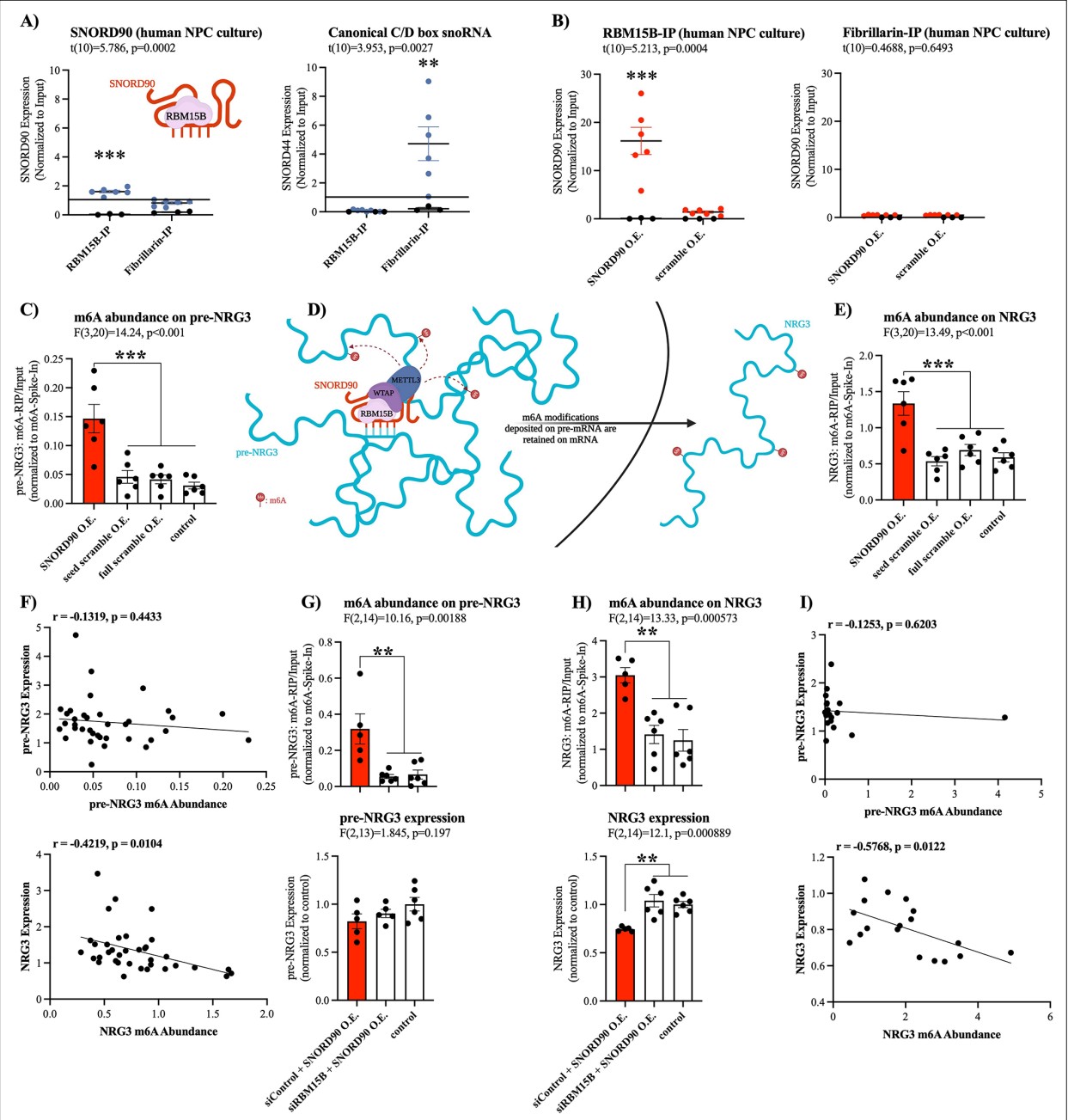

**Figure 4.** *SNORD90* is a guide RNA for RBM15B and increases m6A abundance on *NRG3*. (**A**) Presence of *SNORD90* (left) and canonical snoRNA, *SNORD44*, (right) in RBM15B-IP (n=6) and fibrillarin-IP (n=6) fractions. *SNORD90* displayed higher levels in association with RBM15B as compared to fibrillarin, whereas the canonical snoRNA control displayed higher association to fibrillarin and almost no association with RBM15B. Black dots are IgG negative control (n=3) (**B**) Presence of *SNORD90* in RBM15B-IP (left) and fibrillarin-IP (right) following transfection of *SNORD90* (SNORD90 O.E.; n=6) or scramble (scramble O.E.; n=6) expression vectors in human NPC culture. SNORD90 O.E. increased *SNORD90* detection after RBM15B-IP whereas detection of *SNORD90* in fibrillarin-IP remained unchanged, suggesting that *SNORD90* has preferential binding to RBM15B as compared to fibrillarin. Black dots indicate IgG negative control (n=3) (**C**) Abundance of m6A modifications on *pre-NRG3* following transfection of *SNORD90* (SNORD90 O.E.; n=6), seed scramble (seed scramble O.E.; n=6), and full scramble (full scramble O.E.; n=6) expression vectors in human NPC culture; non-transfected NPCs under normal culture conditions as controls (control; n=6) (**D**) Schematic diagram of *SNORD90's* role in guiding m6A-methyltransferase complex onto *pre-NRG3* in the nucleus. M6A modifications deposited onto *pre-NRG3* are retained on mature *NRG3*. (**E**) Same as (**C**) but for mature *NRG3*. (**F**) Correlation between m6A abundance and RNA expression for *pre-NRG3* (top) and mature *NRG3* (bottom). Significant correlation only observed for mature *NRG3*. (**G–H**) Human NPCs were first transfected with dsiRNA to knock-down RBM15B (siRBM15B) or scrambled control (siControl) followed by transfection with SNORD90 expression vector (siRBM15B+SNORD90 O.E., n=5; siControl +SNORD90 O.E., n=6). Non-transfected NPCs under normal culture conditions as controls (control, n=6). M6A abundance (top) and RNA expression (bottom) were measured for *pre-NRG3* (**G**) and mature

*Figure 4 continued on next page*

*Figure 4 continued*

*NRG3* (**H**). (**I**) Correlation between m6A abundance and RNA expression for *pre-NRG3* (top) and mature *NRG3* (bottom). Correlation data from (**G–H**). Significant correlation only observed for mature *NRG3*. All bar plots represent the mean with individual data points as dots. Error bars represent S.E.M. Student's two-tailed t test (**A–B**). Pearson correlation (**F**) & (**I**).One-way ANOVA with Bonferroni post-hoc (**C & E & G-H**). **$p < 0.01$, ***$p < 0.001$.

The online version of this article includes the following source data and figure supplement(s) for figure 4:

**Figure supplement 1.** m6A abundance and RBM15B knock-down in human NPC culture.

**Figure supplement 1—source data 1.** Original uncropped western blots for RBM15B and GAPDH found in *Figure 4—figure supplement 1C*.

over-expression did not alter m6A abundance (*Figure 4C–E*). Furthermore, introducing target blockers blunted the increase of m6A abundance on both *NRG3* and *pre-NRG3* (*Figure 4—figure supplement 1A–B*). Interestingly, we observed a significant negative correlation between m6A levels and expression of *NRG3*, but this was not observed for *pre-NRG3*, which could possibly be explained by the fact that m6A-readers are primarily located in the cytoplasm, and thus would only recognize the m6A-modifications of the mature *NRG3* transcript and promote its decay when it is shuttled out of the nucleus into the cytoplasm (*Figure 4D and F*). *Ke et al., 2017* demonstrated that m6A modifications are added onto nascent pre-mRNA during transcription (*Ke et al., 2017*). Moreover, m6A levels remain unchanged between nascent pre-mRNA and steady-state mRNA in the cytoplasm (*Ke et al., 2017*). To further test RBM15B's role in the observed increase in m6A levels on *pre-NRG3* and *NRG3*, we used dicer-substrate short interfering RNAs (dsiRNAs) to knock-down RBM15B (*Figure 4—figure supplement 1C–D*). RBM15B knock-down followed by *SNORD90* over-expression blunted the increase of m6A levels on *pre-NRG3* and *NRG3* (*Figure 4G–H*). RBM15B knock-down also blunted the decrease of *NRG3* expression following *SNORD90* over-expression while *pre-NRG3* levels were unaffected (*Figure 4G–H*). We again only observed a significant negative correlation between m6A levels and expression of *NRG3* but not *pre-NRG3* (*Figure 4I*). Our results further support the findings from *Ke et al., 2017*, as it is possible that *SNORD90* is guiding m6A modifications onto *NRG3* at the pre-mRNA level, which are retained in the mRNA molecule (*Figure 4D*). Although it is unclear why *SNORD90* targets intronic regions, while it seems that m6A modifications are deposited onto exonic regions, this could perhaps be explained by the secondary structure of the RNA molecule where exonic regions could be looping closer to the *SNORD90*-guided methylation complex (*Figure 4D*).

The YTH family of proteins are the best described readers of RNA m6A modifications. The YTH family includes: YTHDF1, YTHDF2, YTHDF3, YTHDC1, and YTHDC2. Of these, YTHDF1, YTHDF2, and YTHDF3 regulate RNA stability, whereas functional implications for YTHDC1 and YTHDC2 are less well defined (*Meyer and Jaffrey, 2017*; *Zaccara and Jaffrey, 2020*). Furthermore, YTHDF1, YTHDF2, YTHDF3, and YTHDC2 are primarily located in the cytoplasm, whereas YTHDC1 is primarily located in the nucleus (*Zaccara et al., 2019*). We selectively knock-down each of the above-mentioned m6A-readers using dsiRNAs followed by *SNORD90* over-expression to elucidate which m6A-reader(s) are involved in the down-regulation of NRG3 (*Figure 5—figure supplement 1A–J*). Although knocking down YTHDF1, YTHDF2, and YTHDF3 all showed the ability to blunt decreased *NRG3* expression following *SNORD90* over-expression, YTHDF2 knock-down displayed the most robust ability to blunt this effect, recovering *NRG3* expression to wild-type patterns (*Figure 5A*). On the other hand, knock-down of YTHDC2 and YTHDC1 did not blunt the decrease of *NRG3* expression after *SNORD90* over-expression; displaying similar patterns of *NRG3* expression to *SNORD90* over-expression with a dsiRNA scramble control (*Figure 5A*). Interestingly, YTHDC1 knock-down (the nuclear m6A-reader), resulted in a decrease of *pre- NRG3* expression compared to wild-type, while knock-down of all other m6A-readers had no significant effect on *pre-NRG3* expression levels (*Figure 5—figure supplement 1K*). Together this evidence supports the hypothesis that (1) *SNORD90* increases m6A abundance on *pre-NRG3*, which is retained onto mature *NRG3* transcripts and (2) the increase in m6A abundance promotes YTHDF2 mediated *NRG3* decay which occurs primarily in the cytoplasm (*Figure 5B*). This, in turn, explains why we only observed changes in NRG3 mRNA and protein levels, but did not observe changes in pre-mRNA levels following *SNORD90* over-expression.

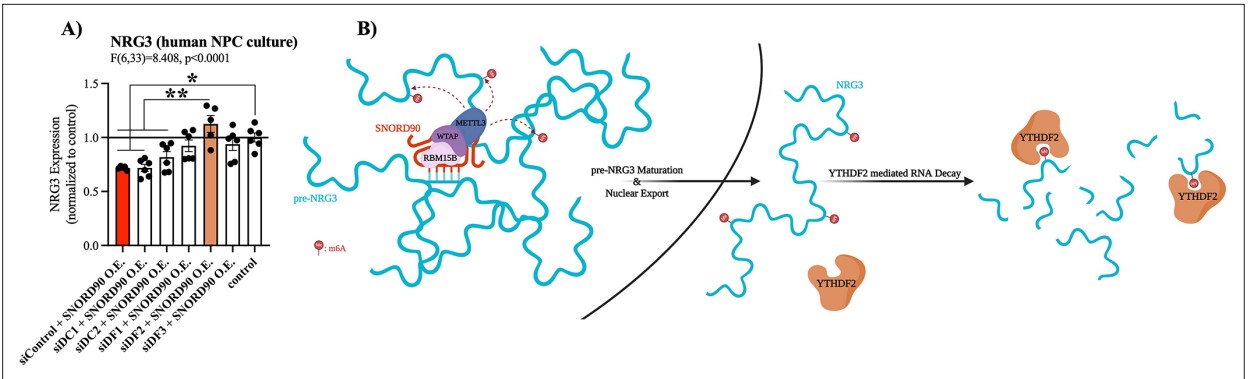

**Figure 5.** *SNORD90*-induced m6A on *NRG3* is recognized by YTHDF2. (**A**) *NRG3* expression in human NPCs transfected with dsiRNA to knock-down the m6A readers YTHDF1 (siDF1), YTHDF2 (siDF2), YTHDF3 (siDF3), YTHDC1 (siDC1), and YTHDC2 (siDC2) or a scramble control (siControl) followed by transfection with SNORD90 expression vector (siDF1 +SNORD90 O.E., n=6; siDF2 +SNORD90 O.E., n=5; siDF3 +SNORD90 O.E., n=6; siDC1 +SNORD90 O.E., n=6; siDC2 +SNORD90 O.E., n=6, siControl +SNORD90 O.E., n=6). Non-transfected NPCs under normal culture conditions as controls (control, n=6). YTHDF2 knock-down showed the most robust ability to blunt *SNORD90's* downregulatory effect on *NRG3* suggesting it plays the largest role in recognizing m6A abundance on *NRG3*. (**B**) Schematic overview of SNORD90's regulation of *NRG3* expression. *SNORD90* interacts with m6A methyltransferase complex in the nucleus and guides this complex onto *pre-NRG3* increasing m6A abundance. The increase in m6A abundance is maintained throughout pre-mRNA maturation (supposedly this increase in m6A is being deposited onto exonic regions) is not recognized until *NRG3* reaches the cytoplasm where it undergoes YTHDF2 mediated RNA decay. Bar plot represent the mean with individual data points as dots. Error bars represent S.E.M. One-way ANOVA with Bonferroni post-hoc. *p<0.05, **p<0.01.

The online version of this article includes the following source data and figure supplement(s) for figure 5:

**Figure supplement 1.** m6A-reader knock-down in human NPC culture.

**Figure supplement 1—source data 1.** Original uncropped western blots for YTHDF1 and GAPDH found in *Figure 5—figure supplement 1A*.

**Figure supplement 1—source data 2.** Original uncropped western blots for YTHDF2 and GAPDH found in *Figure 5—figure supplement 1B*.

**Figure supplement 1—source data 3.** Original uncropped western blots for YTHDF3 and GAPDH found in *Figure 5—figure supplement 1C*.

**Figure supplement 1—source data 4.** Original uncropped western blots for YTHDC1 and GAPDH found in *Figure 5—figure supplement 1D*.

**Figure supplement 1—source data 5.** Original uncropped western blots for YTHDC2 and GAPDH found in *Figure 5—figure supplement 1E*.

## *SNORD90* mediates down-regulation of Nrg3 resulting in increased glutamatergic neurotransmission

We next investigated the functional relevance of decreased NRG3 expression in vivo. Nrg3 has previously been shown to interact with syntaxin, disrupting SNARE complex formation in the presynaptic terminal, and inhibiting vesicle docking in mice (*Figure 6—figure supplement 1*; *Wang et al., 2018*). Nrg3 knock-out resulted in increased probability of glutamate release in mouse hippocampal neurons (*Wang et al., 2018*). To investigate if *SNORD90*-mediated down-regulation of NRG3 has a similar effect in mice, we first investigated if *SNORD90's* ability to down-regulate NRG3 is conserved in mice by in silico target prediction between mouse *Snord90* and *Nrg3* using PLEXY (*Figure 6A*, *Supplementary file 11*). Although predicted sequence complementary sites between *Snord90* and *Nrg3* are not as robust as in humans, there is conservation between the two species (*Figure 6A*). To further explore this effect, we over-expressed *Snord90* or a full scrambled *Snord90* sequence control in the mouse ACC (cg1/2) via bilateral injections of an AAV virus. RT-qPCR confirmed successful over-expression of *Snord90* and subsequent down-regulation of *Nrg3* in mice (*Figure 6B*). Next, we performed whole-cell patch-clamp recordings from pyramidal neurons of the ACC in acute brain slices from mice over-expressing *Snord90* and scramble control. We observed an increase in spontaneous excitatory postsynaptic currents (sEPSC) frequency following *Snord90* over-expression without any effect on sEPSC amplitude (*Figure 5C–E*). This increase in glutamatergic neurotransmission is likely due to an increase in glutamate release probably resulting from Nrg3 pre-synaptic expression and effect on SNARE complex (*Wang et al., 2018*). Together this indicates that *SNORD90*-mediated down-regulation of NRG3 has implications in glutamate neurotransmission, which translates to behavioral changes such as anxiolytic and anti-depressive-like behaviors (*Figure 2*).

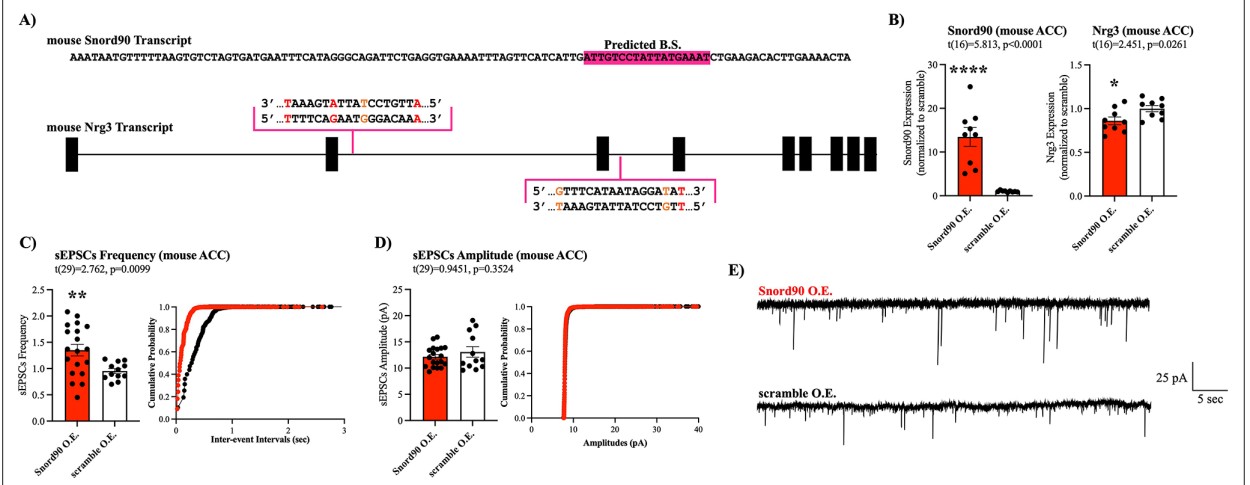

**Figure 6.** *Snord90* induced down-regulation of *Nrg3* increases glutamatergic neurotransmission. (**A**) Full sequence of mature mouse *Snord90* transcript with highlighted region predicted to bind to mouse *Nrg3*. Schematic representation of Nrg3 pre-mRNA transcript indicating regions on *Nrg3* where *Snord90* is predicted to bind (red nucleotide indicates mismatch, orange nucleotide indicates G-T wabble pair). (**B**) Viral injection of *Snord90* expression vector (Snord90 O.E., n=9) or scramble control expression vector (scramble O.E., n=9) into Cg1/2 followed by qPCR confirmation *Snord90* over-expression (left) and subsequent *Nrg3* down-regulation (right). (**C–E**) Whole-cell patch-clamp recordings in Cg1/2 acute brain slices from mice. sEPSCs frequency (**C**) and sEPSCs amplitude (**D**) with cumulative probability plots (Snord90 O.E., n=19 neurons from 8 animals; scramble O.E., n=12 neurons from 3 animals) (**E**) Representative trace recording of sEPSCs in Cg1/2 pyramidal neurons. All bar plots represent the mean with individual data points as dots. Error bars represent S.E.M. Student's two-tailed T-test. *p<0.05, **p<0.01, ****p<0.0001.

The online version of this article includes the following figure supplement(s) for figure 6:

**Figure supplement 1.** A schematic diagram depicting the working model proposed by *Wang et al., 2018* showing NRG3's role in regulating glutamatergic neurotransmission.

## Discussion

In this study, we investigated the expression of snoRNAs in peripheral blood collected across three independent clinical cohorts from patients with MDD that underwent eight weeks of antidepressant treatment. From this investigation our results pointed to an up-regulation of *SNORD90* having a consistent association with antidepressant treatment response across all three of our clinical cohorts. Although other snoRNAs were also identified to have an association with antidepressant response, they were not consistently differentially expressed across all three of our clinical cohorts. Through a series of experiments using rodent models, post-mortem human brain samples, and human neuronal cultures we further determined that the up-regulation of *SNORD90* is primarily achieved by anti-depressant drugs and not by other psychotropic drugs. It is unclear why *SNORD90* seems to be preferentially up-regulated by antidepressant drugs, but we speculate it is due to the mode of action of antidepressant drugs, many of which modulate the serotonergic system. Serotonin signaling via binding to serotonin receptors has been shown to be linked with chromatin remodeling thereby modulating gene expression (*Holloway and González-Maeso, 2015*). Accordingly, a recent study points to the ability of serotonin to modify histones. Specifically, the glutamine at the Q5 site in histone H3 can be modified by serotonin which promotes the recruitment of transcription factor II D (TFIID) and promotes gene expression (*Farrelly et al., 2019*). This process, however, is dependent on intracellular levels of serotonin, more specifically, the levels of serotonin in the nucleus, and it remains unclear if antidepressants can induce such changes.

A combination of in silico target prediction and a series of gain and loss of *SNORD90* function assays indicates that *SNORD90* is able to modulate the expression of mRNA transcripts, particularly *NRG3*. Using additional in silico prediction tools and follow-up wet lab experiments we identified a high confidence sequence motif on *SNORD90* for RBM15B, an essential component of the larger m6A methyltransferase complex (*Van Nostrand et al., 2020*; *Patil et al., 2016*). This is a significant divergence from canonically functioning C/D box snoRNAs that are known to associate with fibrillarin, a methyltransferase responsible for 2'-O-methylation (2'OMe) (*Ideue et al., 2009*; *Newman et al., 2000*). Whereas 2'OMe can occur on the ribose of any base and is associated with increased RNA

stability, m6A is an adenosine specific modification with a diverse range of effects on RNA stability (*Zaccara et al., 2019*; *Dimitrova et al., 2019*). We followed up by measuring m6A abundance on *pre-NRG3* and *NRG3* and found that elevated levels of *SNORD90* is associated with increased m6A abundance on both *pre-NRG3* and *NRG3*. Together our results suggest that SNORD90 is acting as a guide RNA to sequester the m6A-writer complex onto target transcripts in the nucleus. We further determined that the increase of m6A abundance on *NRG3*, as a result of up-regulated *SNORD90* expression, mediates NRG3 mRNA decay via recognition from the m6A-reader YTHDF2. NRG3 is a member of the neuregulin family of epidermal growth factor like signaling molecules involved in cell-to-cell communication (*Avramopoulos, 2018*). There are four known human neuregulin paralogs: NRG1, NRG2, NRG3, and NRG4. NRG1 is the most well studied, however there has been growing interest in NRG3 in neuropsychiatric research. NRG3 was first identified in 1997 and described as an erb-b2 receptor tyrosine kinase 4 (ERBB4) ligand enriched in neural tissue (*Zhang et al., 1997*). Additionally, according to the Genotype-Tissue Expression (GTEx) project, NRG3 is almost exclusively expressed in the central nervous system with the highest expression in the anterior cingulate and frontal cortex; NRG3 also displays the strongest specificity to the central nervous system as compared to the other neuregulin paralogs (*Avramopoulos, 2018*). NRG3 has been associated with several psychiatric illnesses, primarily schizophrenia but also MDD and bipolar disorder; however, the functional relevance of NRG3 remains poorly understood (*Paterson et al., 2017*; *Chen et al., 2009*; *Kao et al., 2010*; *Wang et al., 2008*). A more recent study suggested that NRG3 acts in a cell-autonomous manner specifically in pyramidal neurons and not by activating ERBB4 receptors (*Wang et al., 2018*). They propose that NRG3 interacts with syntaxin 1(specially at the SNARE domain), at the pre-synaptic terminal, resulting in the inhibition of SNARE-complex formation preventing vesicle docking and reducing glutamate release (*Wang et al., 2018*). Furthermore, by utilizing a NRG3 knock-out mouse model, they observed a significant increase in glutamatergic signaling (*Wang et al., 2018*). This finding was of significant interest to us since our data points to a down-regulation of NRG3 mediated by up-regulation of *SNORD90* after antidepressant treatment. To determine if our *SNORD90*-NRG3 regulatory network has implications on glutamatergic signaling, we over-expressed *SNORD90* in the mouse anterior cingulate cortex followed by slice electrophysiology recordings where we observed an increase in spontaneous excitatory post synaptic currents (sEPSC). Future studies concomitantly manipulating levels of *Nrg3* and *Snord90* will be important to further test the effects noted in this study.

In summary, our proposed model suggests that antidepressant treatment is linked to an up-regulation of *SNORD90*, which in turn recruits the m6A-writer complex and guides this complex onto *pre-NRG3* increasing the abundance of m6A modifications which is further retained onto the mature *NRG3* transcript. In turn, the greater abundance of m6A modifications on *NRG3* is recognized by the m6A-reader, YTHDF2, which mediates *NRG3* decay. This reduction in NRG3 levels is associated with an increase in glutamatergic signaling which, we believe, contributes to antidepressant response.

The glutamatergic system has become of great interest in antidepressant therapeutics research since drugs targeting this system, in particular ketamine, results in rapid alleviation of MDD symptoms. It has been repeatedly shown that monoaminergic antidepressants have associative effects on the glutamatergic system, however the exact molecular mechanistic link between the two system has remained unknown (*Musazzi et al., 2013*). Our findings suggest that snoRNAs play a role in antidepressant treatment response and provides a molecular link between monoaminergic targeting antidepressants to implications to the glutamatergic system. Several theories postulate that the therapeutic effects of ketamine and other rapid acting antidepressant is through NMDA receptor inhibition. Among these theories, the disinhibition hypothesis suggests that ketamine may be preferentially inhibiting NMDA receptors located on GABAergic interneurons thus disinhibiting excitatory pyramidal neurons leading to an enhancement of glutamatergic neurotransmission. In contrast, other theories propose that ketamine directly inhibits glutamatergic neurotransmission via NMDA receptor inhibition (*Zanos and Gould, 2018*). Our findings support the notion that the monoamine system may be playing more of a modularly role to the glutamatergic system to achieve an antidepressant outcome; in particular we observed that an increase in glutamatergic neurotransmission is associated with a response to monoaminergic-acting antidepressant.

In addition to the contribution to understanding antidepressant drug action, we also contribute to the growing knowledge pertaining to the functional mechanisms of snoRNAs. Beyond their canonical

regulatory roles, snoRNAs have been reported to influence gene expression via interaction with RNA targets however the mechanism by which this occurs remained unclear. We are the first to propose that snoRNAs play a role in m6A modifications which ultimately has regulatory roles in the stability of mRNA.

## Materials and methods

### Human clinical trial subjects

Three independent cohorts were used in this study, comprising 660 individuals (*Lopez et al., 2017*; *Kennedy et al., 2019*; *Jollant et al., 2020*).

Cohort 1 (N=258) was obtained in collaboration with Lundbeck A/S sponsored clinical trials and is composed of individuals diagnosed with MDD in a current major depressive episode (MDE) who were enrolled in a double-blind clinical trial and received treatment with either duloxetine (60 mg), a serotonin-norepinephrine reuptake inhibitor (SNRI), or placebo for eight weeks. For each patient, peripheral blood samples were collected at baseline (T0) and after treatment (T8). Participants, aged 19–74 years, were recruited based on a primary diagnosis of MDD and MDE lasting at least 3 months, with a severity score on the Montgomery-Åsberg Depression Rating Scale (MADRS) of ≥22 at T0. Participants resistant to at least two previous AD treatments or who had received electroconvulsive therapy in the 6 weeks before the study began were excluded. Other exclusion criteria included: MDE in bipolar disorder, presence of psychotic features, and recent substance use disorder. This clinical trial was approved by ethics boards of participating centers, and all participants provided written informed consent. https://www.clinicaltrials.gov/ (11984 A NCT00635219; 11918 A NCT00599911; 13267 A NCT01140906). For more details, please refer methods section of *Lopez et al., 2017*.

Cohort 2 (N=236) was obtained in collaboration with the Canadian Biomarker Integration Network in Depression (CAN-BIND) and is composed of individuals diagnosed with MDD (N=153) and healthy controls (N=83). Patients and controls were recruited at six Canadian clinical centers. Exclusion criteria included personal and family history of schizophrenia or bipolar disorder, or current substance dependence. Depressed patients were treated with escitalopram (10–20 mg per day), a selective serotonin reuptake inhibitor (SSRI), for eight weeks. Depression severity was assessed at baseline and after treatment by MADRS. This trial was approved by ethics boards of participating centers and all participants provided written informed consent. https://www.clinicaltrials.gov/ identifier NCT01655706. Registered 27 July 2012. For more details, please refer to *Kennedy et al., 2019*.

Cohort 3 (N=166) was obtained at the Douglas Mental Health University Institute and is composed of healthy controls (N=28) and individuals diagnosed with MDD (N=138) who were enrolled in the community outpatient clinic at the Douglas Mental Health University Institute. Depressed patients were treated with either desvenlafaxine, a serotonin-norepinephrine reuptake inhibitor (SNRI) or escitalopram, a selective serotonin reuptake inhibitor (SSRI). For each patient, a blood sample was taken at baseline before treatment administration and 8 weeks post-treatment. Participants (healthy controls and individuals with MDD) were excluded from the study if they had comorbidity with other major psychiatric disorders, if they had positive tests for illicit drugs at any point during the study, or if they had general medical illnesses. Individuals with MDD were not receiving antidepressant treatment at the onset of the trial, and received a diagnosis of MDD without psychotic features, according to the Statistical Manual of Mental Disorders, Fourth Edition (DSM-IV). Control subjects were excluded if they had a history of antidepressant treatment. Eligible participants were randomized to either desvenlafaxine (50–100 mg) or escitalopram (10–20 mg) treatment. All subjects included in the study provided informed consent, and the project was approved by The Institutional Review Board of the Douglas Mental Health University Institute. For more details, please refer to *Jollant et al., 2020*.

### Clinical assessment

All participants from all cohorts were assessed for depression severity after 6–8 weeks of treatment. To quantify treatment response, we calculated percentage change of MADRS scores (from baseline to after treatment). We used percentage change to correct for the potential effects of differential baseline scores. Additionally, we classified participants as responder/non-responder based on >50% decrease in MADRS scores from baseline.

## Human peripheral blood sample processing and RNA extraction

Peripheral blood samples from cohort 1 were collected in PAXgene blood RNA tubes (PreAnalytix). Total RNA was isolated from whole blood using the PAXgene Blood miRNA Kit (Qiagen, Canada) according to the manufacturer's instructions. Peripheral blood samples from cohort 2 and cohort 3 were collected in EDTA blood collection tubes and passed through LeukoLOCK filters (ThermoFisher) to capture the total leukocyte population, eliminating red blood cells, platelets, and plasma. Filters, containing leukocytes, were frozen at –80 °C for storage until ready for sample processing. Total RNA was extracted using a modified version of the LeukoLOCK Total RNA Isolation System protocol (ThermoFisher). All samples were treated with DNase digestion during RNA purification using the RNase-Free DNase kit (Qiagen). RNA yield and quality were determined using the Nanodrop 1000 (Thermo Scientific, USA) and Agilent 2200 Tapestation (Agilent Technologies, USA).

## Library construction and small RNA-sequencing

Libraries for cohort 1 and cohort 3 were prepared using the Illumina TruSeq Small RNA protocol following the manufacturer's instructions. Libraries for cohort 2 were prepared using NEB small RNA protocol following manufacturer's instructions. All libraries were purified using biotinylated magnetic AMPure beads that allow for selection of specified complementary cDNA products bound to streptavidin. A total of 50 µl of amplified cDNA were mixed and purified twice with AMPure XP beads in a 1.8:1 ratio (beads/sample). Cohort 1 was sequenced using Illumina HiSeq2500, cohort 2 was sequenced using Illumina HiSeq4000, and cohort 3 was sequenced using Illumina HiSeq2000. All samples were sequenced at the McGill University and Genome Quebec Innovation Centre (Montreal, Canada) using 50 nucleotide single-end reads. All sequencing data was extracted from FASTQ files and processed using CASAVA 1.8+. Illumina adapter sequences were trimmed using the Fastx_toolkit, and additionally filtered by applying the following cut-offs: (*Brody and Gu, 2020*) Phred quality (Q) score higher than 30, (*Cipriani et al., 2018*) reads between 15-40nt in length (*Harmer et al., 2017*), adapter detection based on perfect-10nt match, and (*Ross and Renyi, 1969*) removing reads without detected adapters. Bowtie35 (John Hopkins University) was used to align reads to the human genome (GRCh37). Furthermore, Rfam database was used to map reads to known small nucleolar RNAs. Sequencing data was normalized with the Bioconductor-DESeq2 package.

## Statistical analysis

For each subject in these trials, samples collected before administration of an antidepressant (T0) and 8 weeks following antidepressant treatment (T8) were analyzed according to response to treatment based on MADRS score changes. For the discovery cohort, antidepressant treated, and placebo treated subjects were analyzed separately.

A two-way mixed multivariable analysis of variance (2WM-MANOVA) was used to identify snoRNAs that had a significant interaction between treatment response (response/non-response) and treatment course (T0/T8). For cohort 1, antidepressant treated, and placebo treated subjects were analyzed separately. All detected snoRNAs were assessed in cohort 1. Only snoRNAs that showed significant interactions were assessed in cohort 2 and subsequently only snoRNAs that were replicated in cohort 2 were assessed in cohort 3. For all cohorts, outliers were identified using boxplot methods with values above quartile 3 (Q3) +1.5 interquartile range (IQR) or below quartile 1(Q1) - 1.5IQR (IQR = Q3-Q1). Shapiro-Wilk test and QQ plots were used to assess data normality. All above mentioned tools were apart of publicly available R package 'rstatix' (https://CRAN.R-project.org/package=rstatix) (*Kassambara, 2020*).

## Unpredicted chronic mild stress (UCMS) mouse model

All experiments on mice were carried out according to policies on the care and use of laboratory animals of European Community legislation 2010/63/EU. The local Ethics Committee (CEEAVdL-19) approved the protocols used in this study (protocol number 2011-06-10). Eight-week old male BALB/c mice (N=23; Centre d'Elevage Janvier, Le Genest St. Isle, France) were divided into four groups as described by *Hervé et al., 2017* (*Hervé et al., 2017*). In brief, control group (control) was kept in standard housing conditions for 8 weeks. UCMS only group comprised of mice subjected to the Unpredictable Chronic Mild Stress (UCMS) procedure for 8 weeks. UCMS + flx group included mice that were subjected to the UCMS procedure for 8 weeks and treated in parallel by fluoxetine (flx)

during the last 6 weeks. Control +flx group included mice that did not undergo the UCMS procedure but were treated with fluoxetine during the last 6 weeks. Mice from the control and control +flx groups were housed in standard cages, whereas the UCMS-exposed mice were isolated in individual home cages with no physical contact with other mice. The stressors used were varied and applied in a different sequence each week to avoid habituation.

Stressors consisted of housing on damp sawdust (about 200 mL of water for 100 g of sawdust), sawdust changing (replacement of the soiled sawdust by an equivalent volume of new sawdust), placement in an empty cage (usually the home cage of the subject, but with no sawdust), placement in an empty cage with water (the mouse is placed in its empty cage, whose bottom has been filled up with 1 cm high water at 21 °C), switching cages (also sometimes termed as social stress: the mouse from a cage A is placed in the soiled cage from mouse B, mouse B itself being absent in order to avoid aggressive interactions), cage tilting (45°), predator sounds, introduction of rat or cats feces as well as fur in the mouse home cage, inversion of the light/dark cycle, lights on for a short time during the dark phase or light off during the light phase, confinement in small tubes (diameter: 4 cm; length: 5 cm).

### Mice behavior

Weight and coat state were measured weekly, as markers of UCMS-induced depressive-like behavior, except for the last week before sacrifice, when coat state from seven different areas of the body was recorded twice, separated by 3-day intervals. At the end of the 8th week, a complementary test of nest building was performed just before sacrifice. The test was administered by isolating mice in their home cages. For additional details please refer to *Hervé et al., 2017*.

### Quantification and statistical analysis

Details related brain dissection and RNA extraction can be found at *Hervé et al., 2017*. RNA was reverse-transcribed using M-MLV Reverse Transcriptase (200 U/µL) (ThermoFisher) with random hexamers. *Snord90* and *Nrg3* were quantified by RT-PCR using SYBR green (Applied Biosystems). Reactions were run in triplicate using the QuantStudio 6 Flex System and data collected using QuantStudio Real-Time PCR Software v1.3. See *Supplementary file 12* for a list of primer sequences used. One-way ANOVA was used as described above.

## Human post-mortem brain

Post-mortem samples of dorsal ACC (Brodmann Area 24) were obtained, in collaboration with the Quebec Coroner's Office, from the Douglas-Bell Canada Brain Bank (Douglas Mental Health University Institute, Montreal, Quebec, Canada). Groups were matched for post-mortem interval (PMI), pH and age. Psychological autopsies were performed as described previously, based on DSM-IV criteria (*Dumais et al., 2005*). The control group had no history of major psychiatric disorders. All cases met criteria for MDD or depressive disorder not-otherwise-specified. Written informed consent was obtained from next-of-kin. This study was approved by the Douglas Hospital Research Centre institutional review board.

### RNA-sequencing

RNA was extracted from all brain samples using a combination of the miRNeasy Mini kit and the RNeasy MinElute Cleanup kit (Qiagen), with DNase treatment, and divided into small (<200 nt) and large (>200 nt) fractions. RNA quality, represented as RNA Integrity Number, was assessed using the Agilent 2200 Tapestation. Small RNA-seq libraries were prepared from the small RNA fraction, using the Illumina TruSeq Small RNA protocol following the manufacturer's instructions. Samples were sequenced at the McGill University and Genome Quebec Innovation Centre (Montreal, Canada) using the Illumina HiSeq2000 with 50nt single-end reads. All sequencing data were processed using CASAVA 1.8 + (Illumina) and extracted from FASTQ files. The Fastx_toolkit was used to trim the Illumina adapter sequences. Additional filtering based on defined cutoffs was applied, including: (*Brody and Gu, 2020*) Phred quality (Q) mean scores higher than 30 (*Cipriani et al., 2018*), reads between 15 and 40 nt in length (*Harmer et al., 2017*), adapter detection based on perfect-10nt match, and (*Ross and Renyi, 1969*) removal of reads without detected adapter. In addition, we used Bowtie [24] to align reads to the human genome (GRCh37). Furthermore, all sequencing data was normalized with the Bioconductor—DESeq2 package, using a detection threshold of 10 counts per snoRNA. We retained

all snoRNAs with >10 reads in 70% of either group (controls, cases) for differential analyses. RNA extractions, sequencing, and data processing were conducted by blinded investigators. For additional details please refer to *Fiori et al., 2021*.

## Statistical analysis

Using toxicology screens, each sample was separated into the following groups: MDD with presence of antidepressants (MDD +AD), MDD with presence of non-antidepressant drugs (MDD +nonAD drugs), MDD with negative toxicology screen (MDD), controls with non-antidepressant drugs (control +nonAD drugs), and controls with negative toxicology screens (control). No control samples were positive for antidepressant drugs. For this analysis we only investigated the expression for *SNORD90* from this small RNA sequencing dataset. *NRG3* was measured via qPCR from cDNA converted from the same RNA aliquot for each sample. See *Supplementary file 12* for a list of primer sequences used. Analysis was performed in R using "rstatix" as mentioned above employing a one-way ANOVA. Shapiro-Wilk test and QQ plots were used to assess data normality.

## Fluorescence-activated nuclei sorting

### Nuclear extraction and labeling

To purify intact nuclei from human post-mortem samples, we homogenized 50 mg of frozen tissue in nuclei buffer containing 10 mM PIPES (pH 7.4), 10 mM KCl, 2 mM MgCl2, 1 mM dithiothreitol (DTT), and 10×Protease Inhibitor Cocktail (Sigma-Aldrich). Homogenates were passed through a 30% sucrose gradient in nuclei buffer in order to separate nuclei from cellular debris, then after a wash with nuclei buffer, nuclei pellets were resuspended in blocking buffer containing 0.5% bovine serum albumin (BSA) in 10×normal goat serum. Each sample was incubated with anti-NeuN-PE (1:300) (Millipore, FCMAB317PE) for 60 min at 4 °C in the dark. DRAQ5 (1:300) (Thermo Fisher) was added to each sample and gently agitated before moving to nuclei sorting step.

### Nuclei sorting and RNA extraction

Labeled nuclei were passed through 40 μM filter caps to remove any remaining cellular debris and processed using the BDFACSAria III platform (BD Biosciences) according to technical specifications provided by the company. We used BD FACSDIVA Software (BD Biosciences) to first isolate single, intact nuclei based on DRAQ5 fluorescence at the 730/45 A filter (DRAQ5), and then to sort neuronal from non-neuronal nuclei based on fluorescence detected by the 585/42 filter (PE). Sorted nuclear fractions were collected in sheath fluid (1×phosphate-buffered saline (PBS)). Upon sort completion, 3 X volume Qiazol was immediately added to each fraction and stored at –80 °C until RNA extraction. RNA was extracted using Direct-zol RNA extraction kit (Zymo) following manufacturer instructions. Due to larger volume of starting material a vacuum system was used instead of centrifugation for the initial RNA extraction step. cDNA conversion and *SNORD90* quantification was same as detailed above with slight modifications. Instead of random hexamers, *SNORD90* reverse primer (40 μM) was used in cDNA construction to aid in *SNORD90* quantification via qPCR.

## Human hindbrain NPC culture and neuronal differentiation

Monoamine producing neurons were generated from human induced pluripotent stem cells (iPSCs), using a protocol adapted from *Lu et al., 2016*. Human iPSCs were first cultured in DMEM/F12 (Gibco) supplemented with N2 (Gibco), B27 (Gibco), nonessential amino acids (Gibco), 1% GlutaMAX (Gibco), 2 μM SB431542 (STEMCELL Tech.), 2 μM DMH1 (Tocris), and 3 μM CHIR99021 (Tocris); collectively referred to as SDC media. Culturing in SDC media for 1 week induces human iPSC differentiation into rostral hindbrain neural stem cells (NSCs). Rostral hindbrain NSCs colonies were selected and re-plated in SDC media supplemented with 1000 ng/ml of SHH C25II (GenScript). Hindbrain NPCs were *GBX2*, *HOXA2*, and *HOXA4* positive as assessed via quantitative RT-PCR to confirm hindbrain specificity at this developmental stage. Ventral rostral hindbrain NSC colonies were collected and re-plated in SDC +SHH media along with 10 ng/ml of FGF4 (PeproTech). SDC +SHH + FGF4 media will induce ventral rostral hindbrain NSC differentiation into neural progenitor cells (NPCs) after 1 week. NPCs were expanded in SDC +SHH + FGF4 media and cryopreserved (P0). All experiments thereafter used NPCs generated by this protocol. Mycoplasma contamination was tested via qPCR using media the NPCs were cultured in; no contamination was detected.

NPCs were differentiated, for 1 month, into neuron-like cells in neurobasal media (Gibco) supplemented with N2, B27, NEAA, 1 ug/ml laminin (Sigma), 0.2 mM vitamin C (Sigma), 2.5 uM DAPT (Sigma), 10 ng/ml GDNF (GenScript), 10 ng/ml BDNF (GenScript), 10 ng/ml insulin-like growth factor-I (Pepro Tech), and 1 ng/ml transforming growth factor β3 (Pepro Tech). Post-differentiation, neuron-like cells underwent high-performance liquid chromatography (HPLC) to test for monoamine production. HPLC confirmed production of norepinephrine (NE), epinephrine (epi), dopamine (DA), and serotonin (5-HT). All NPCs and differentiated NPCs were seeded on culture plates coated with 100 μg/mL poly-L-ornithine (sigma) and 10 μg/mL laminin (Sigma) and grown in a 5% $CO_2$ humidified incubator at 37 °C.

## High-performance liquid chromatography

Cells were sonicated in 60 μL of 0.25 N perchloric acid. Protein and cellular debris were cleared by centrifugation at 11,000 *g* at 4 °C for 10 min. Pellets were re-suspended in 100 μl of 0.1 N NaOH. Twenty μL of tissue homogenates, cleared of protein and debris, was injected using a refrigerated ultiMate 3000 rapid separation autosampler (ThermoFisher) into a HPLC system consisting of a luna 3 u C18 (*Cipriani et al., 2018*) 100 A 75X4.6 mm phenomemex and a coulometric electrochemical detector (ThermoFisher) to quantify monoamines. Oxidation and reduction electrode potentials of the analytical cell (5014B; ThermoFisher) were set to +300 and −250 mV respectively. The mobile phase consisting of 73.4 mM sodium acetate trihydrate, 66.6 mM of citric acid monohydrate, 0.025 mM Na2EDTA, 0.341 mM 1-octanesulfonic acid, 0.71 mM of Triethylamine and 6% (v/v) methanol (pH adjusted to 4.0–4.1 with acetic acid) was pumped at 1.5 ml/min by a solvent delivery module (dionex ultimte 3000 rs pump).

## NPC drug treatment

Human NPCs were screened for cytotoxic effects using the MTT assay, and antidepressants were applied at nontoxic concentrations as described by *Lopez et al., 2014*. NPCs were cultured in 24-well plates and differentiated, for 2 weeks, into neuron-like cells in neurobasal media (Gibco) supplemented with N2, B27, NEAA, 1 μg/ml laminin (Sigma), 0.2 mM vitamin C (Sigma), 2.5 μM DAPT (Sigma), 10 ng/ml GDNF (GenScript), 10 ng/ml BDNF (GenScript), 10 ng/ml insulin-like growth factor-I (Pepro Tech), and 1 ng/ml transforming growth factor β3 (Pepro Tech). Following 2 weeks of differentiation, culture media was supplemented with escitalopram (Sigma-Aldrich, E4786; 100 μM), duloxetine (Sigma-Aldrich, Y0001453; 10 μM), haloperidol (Sigma-Aldrich, H1512; 10 μM), lithium (Sigma-Aldrich, L4408; 1 mM), Aspirin (Sigma-Aldrich, A5376; 1 mM), or left untreated (controls). Cells for each drug treatment were incubated for 48 h before harvest and RNA extractions. Each drug treatment was performed in triplicate. RNA was extracted using the Zymo DirectZol RNA Extraction kit. cDNA construction and RT-qPCR were as described above. One-way ANOVA analysis was performed in IBM SPSS Statistics version 27 using Dunnett's post-hoc correction.

## *Snord90* over-expression in mice

All experiments were performed in accordance with the European Communities' Council Directive 2010/63/EU. All protocols were approved by the Ethics Committee for the Care and Use of Laboratory Animals of the government of Upper Bavaria, Germany and by the Institutional Animal Care and Use Committee (IACUC) of the Weizmann Institute of Science (Rehovot, Israel). Male CD-1 (ICR) were housed in temperature-controlled (23 ± 1°C), constant humidity (55 ± 10%), 12 hr light/dark cycle and in specific-pathogen-free conditions. Animals had access to food and water *ad libidum*. Animals were housed in groups of four.

## Cloning

The *Snord90* sequence was obtained from the Ensembl genome browser database (Ensemble ID ENSMUSG00000077756) (*Supplementary file 4*). The control sequence was generated by scrambling the *Snord90* gene using siRNA wizard software (Invitrogen) (*Supplementary file 4*). Each fragment was designed to have KpnI and BamHI and a 10-nucleotide long overhang with the vector backbone at the 5' and 3' ends respectively. Each gene fragment was inserted into a KpnI with BamHI linearized pAAV-EF1a-eGFP-H1 backbone using Gibson Assembly (NEB) according to the manufacturer's

protocol. This generated the following two vectors, pAAV-EF1a-eGFP-H1-Scramble and pAAV-EF1a-eGFP-H1-Snord90. All plasmids were checked for mutations by DNA sequencing.

## Validation of constructs

Mouse neuroblastoma neuro2a (N2a) cells were maintained at 37 °C with 5% $CO_2$ in Minimum Essential Medium (MEM), 1 x Glutamax, supplemented with 1 x non-essential amino acids, 1 mM sodium pyruvate, 100 U/ml penicillin, 100 µg/ml streptomycin and 10% fetal bovine serum (FBS, Gibco). Cells were detached with trypsin and transfected using ScreenfectA (ScreenFect GmbH) according to the manufacturer's protocol. Cells were fixed with 4% PFA-PBS solution and embedded with Fluoromount-G mounting medium containing DAPI (SouthernBiotech). Cells were imaged using an Axioplan 2 fluorescent microscope (Zeiss).

## Virus production

Human embryonic kidney cells (HEK293) were cultured in Dulbecco's Modified Eagle Medium (DMEM) supplemented with 10% FBS, 100 U/ml penicillin and 100 µg/ml streptomycin (Invitrogen) in a 5% $CO_2$ humidified incubator at 37 °C. Cells were transfected with the gene transfer rAAV plasmid combined with the helper plasmids in an equal molar ratio of 1:1:1 using 1 mg/ml linear polyethylenimine hydrochloride (PEI). The rAAV (serotype 1/2) particles were harvested three days after transfection by lysing the cells with three consecutive freeze-and-thaw cycles using an ethanol on dry ice bath and 37 °C water bath. Lysates were centrifuged (3000 rcf) followed by purification of the rAAV particles using a Heparin Agarose Type I chromatography column (Sigma). The eluted rAAV particles were PBS washed using a 100000 MWCO Amicon Ultra Filter (Millipore) and suspended in a final volume of 100 µl. The number of viral genomic particles was determined using quantitative RT-PCR resulting in the following titers; AAV1/2-EF1a-eGFP-H1-Scramble $1,67\times10^{11}$ genome particles (gp)/µl and AAV1/2-EF1a-eGFP-H1-Scramble $5\times10^{10}$ gp/µl.

## Stereotactic surgery

Eight-week-old mice were anesthetized with isoflurane and placed in the stereotactic apparatus (TSE Systems) on a 37 °C heating pad. Pre-surgery, mice were given Novalgin (200 mg/kg body weight) and Metacam (sub-cutaneous 0.5 mg/kg body weight). During surgery, mice were continuously supplied with 2% v/v isoflurane in $O_2$ through inhalation. Viruses were injected bilaterally using a 33-gauge injection needle with a 5 µl Hamilton syringe coupled to an automated microinjection pump (World Precision Instruments). Virus was delivered at a rate of 0.1 µl/min, to inject 0.25 µl for behavior experiments, and 0.5 µl for molecular experiments. The injection coordinates were determined using the Franklin and Paxinos mouse brain atlas, from bregma: ML +/-0.3 mm bilateral; AP +1.2 mm; DV −1.8 mm. After injection the needle was retracted 0.01 mm and kept at the site for 2.5 min, followed by slow withdrawal. In all experiments, each group consisted of two scramble control virus injected and two Snord90 overexpression virus injected animals. After surgery, the animals received Metacam for the 3 following days (intraperitoneal 0.5 mg/kg body weight). In all experiments, mice were tested or tissue was extracted 3–4 weeks after surgery. After completion of the experiments, mice were sacrificed by isoflurane overdose. For imaging of brain material, the brains were removed and fixed in 4% PFA-PBS followed by dehydration in 30% sucrose-PBS solution for at least 24 hr each. Brains were sectioned (50 µm) using a vibratome (HM 650 V, Thermo Fisher Scientific). Brain slices were imaged using the VS120-S6-W slide scanner microscope (Olympus). Injection sites were verified based on green fluorescent protein (GFP) expression. For RNA extraction, the brains were extracted and snap-frozen using methylbutane. Brains were sectioned in 200-µm-thick slices in a cryostat and the injection site was collected using 0.8-mm-thick puncher. The tissue was stored at −80 °C until RNA extraction.

## Real-time PCR (RT-PCR)

Total RNA was extracted from cells or tissue using the miRNeasy Mini Kit (Qiagen). cDNA was generated using the high-capacity cDNA RT kit with RNase inhibitor (Applied Biosystems) according to the supplied protocol. RT-PCR was performed according to the manufacturer's instructions using the QuantiFast SYBR green PCR Kit (Qiagen). RT-PCR data was collected on the QuantStudio 7 Flex

Real-Time PCR System (Applied Biosystems). Absolute expression differences were calculated using the standard curve method. See *Supplementary file 12* for a list of primer sequences used.

## Behavioral tests

For all behavioral tests, animals were brought into the room 30 min prior to the start of the test to habituate the animals to the test room.

Open field (OF) test: The OF test was performed in a 50x50 cm light grey box, evenly illuminated with low light conditions (<10 lux). Mice were placed in the open field facing one of the walls and recorded for 6 min. Animals were tracked using ANY-maze software (Stoelting). Total distance traveled was used as measure of animal locomotion.

Elevated plus maze (EPM): EPM apparatus was made of light grey material, and consists of four intersecting arms elevated approximately 30 cm above the floor. The two opposing open (27.5x5 cm) and closed arms (27.5x5 x 20 cm) are connected with a central zone (5x5 cm). The animals were placed in the center of the EPM, facing one of the open arms and recorded for 6 min. The closed arms were illuminated with <10 lux, while the open arms were illuminated with 25–30 lux. Recordings were tracked and analyzed with ANYmaze software. Total time spent in the open or closed arms was used as an anxiety measure.

Splash test (Spl): Animals were placed in a novel cage containing fresh bedding material and were allowed to explore for 5 min. Each animal received two sprays of room temperature 10% sucrose water at the rear of their body. Animals were recorded for 6 min under low-light conditions (<10 lux). The total groom time within the 3–6 min timeframe was manually scored using Solomon Coder software.

Tail suspension test (TST): In the TST, animals were taped by their tales on a metal rod, approximately 30 cm above the ground, and illuminated with 30–35 lux. Animals were recorded for 6 min and struggle time was quantified using ANY-maze software. The total time an animal struggled within the 3–6 min timeframe was used as a measure of a depression-like emotional state.

Z-scoring was used to integrate multiple behavioral tests as previously described (*Guilloux et al., 2011*). First, the z-score of each individual behavioral parameter was calculated.

$$z = \frac{X - \mu}{\sigma}$$

The z-scores of the EPM open arm, EPM closed arm, Spl and TST were combined to calculate an integrated emotionality score.

$$Emotion\,z\,score = \frac{ZEPM + ZSpl + ZTST}{Number\,of\,tests}$$

The final integrated score is associated with anxiety and depressive-like behaviors with a higher score indicating a higher emotional state, while a lower score indicating a lower emotional state.

## Target prediction

The entire sequence of *SNORD90* was used to blast against the human genome allowing G-T wobble base pairing and one mismatch base pairing (*Supplementary file 5*). From this prediction, we identified 100 locations on the human genome (both within genes and gene desert regions) that has complementarity sequences to *SNORD90*. *NRG3* was ranked as one of highest genes housing a complementarity sequence to *SNORD90* without any mismatches. To increase our confidence in target selection, we also utilized the C/D box snoRNA target prediction algorithm PLEXY, using default parameters (*Kehr et al., 2011*). *SNORD90* mature sequence was used as the input snoRNA and target sequence input was the entire human genome downloaded from the Ensembl genome browser database (*Supplementary files 6-7*). Prediction was conducted using whole transcriptomics (*Supplementary file 6*) as well as a targeted analysis for *NRG3* (*Supplementary file 7*). Using the PLEXY algorithm also identified *NRG3* as a possible target for *SNORD90* (*Supplementary file 6*). Although *NRG3* was not ranked among the highest predicted targets using the PLEXY algorithm, *NRG3* was selected as the candidate gene target since it was the only gene that displayed overlap between the two target prediction methods (*Supplementary files 5 and 6*). PLEXY further reviled that *SNORD90* has multiple possible interaction sites on *NRG3*, thus we selected the top three predicted sites for consideration in future experiments (*Figure 3A*, *Supplementary files 6 and 7*).

### *SNORD90* over-expression in human NPC culture

Constructs and cloning were as described above but replacing mouse *Snord90* sequence with human *SNORD90* sequence (*Figure 3—figure supplement 1A* & *Supplementary file 4*). NPCs were plated in 24-well plates until ~90% confluent before being transfected with each vector using Lipofect-amine 2000 (ThermoFisher) following manufacturer's protocol. NPCs were incubated for 48 hr before harvesting for RNA extraction. RNA extraction, cDNA synthesis, and RT-qPCR were as described above. One-way ANOVA was used as described above.

### Target blockers

Modified 2′-O-methoxy-ethyl modified (2MOE) RNA bases with phosphorothiate linkage oligos (target blockers) were designed against three regions on the *NRG3* transcript where *SNORD90* is predicted to bind (*Figure 3A and D* & *Supplementary file 9*). Target blockers were co-transfected (30 nM final concentration) with *SNORD90* expression vectors using lipofectamine 2000 following manufacturer recommendations for 'plasmid DNA and siRNA' co-transfection. For groups where target blockers were pooled together, equal amounts for each target blocker were used totaling 30 nM final concentration. NPC culture conditions, RNA extraction, cDNA synthesis, and RT-qPCR were as described above. One-way ANOVA was used as described above.

### Western blot

Cell pellets from cell culture were lysed in lysis buffer [150 mM NaCl, 50 mM HEPES (pH 7), 50 mM EDTA, and 0.1% NP-40] and quantified using Pierce BCA Protein Assay Kit (Thermo Fisher). Equal amounts of proteins were electrophoresed on 4–20% Mini-PROTEAN TGX Stain-Free Gels (Bio Rad). Proteins were then transferred onto nitrocellulose membranes using the Trans-Blot Turbo Transfer System (Bio Rad). The membranes were blocked with 5% Bovine Serum Albumin (BSA) in Phosphate-Buffered Saline (0.05% Tween 20) (PBS-T) at room temperature for 1–2 hr and then incubated with primary antibody (NRG3 (Abcam, ab109256) at 1:500 and GAPDH (NEB, 3683 S) at 1:1000) in 1% BSA in PBS-T overnight at 4 °C. They were then incubated with either biotin-conjugated anti-rabbit antibody (Vector Laboratories, BA-1000) or with Horseradish Peroxidase (HRP)-conjugated anti-rabbit antibody diluted 1:5000 in 1% BSA in PBS-T for 1 hr at room temperature. The membranes that were incubated with biotin-conjugated anti-rabbit antibody were washed with PBS-T and incubated with streptavidin-conjugated HRP (Jackson ImmunoResearch, 016-030-084) at 1:5000 in 1% NFDM in PBS-T for 1 hr at room temperature. After washing, immunoreactivity was detected using enhanced chemiluminescence solutions (ECL) and the Biorad ChemiDoc MP imaging system.

### *SNORD90* knock-down in human NPC culture

*SNORD90* was knocked-down using antisense oligonucleotides (ASO) containing 2′-O-methylations and phosphorothioate-modified nucleotides (*Supplementary file 8*). Four ASOs and one scrambled control ASO were screened to identify which achieved the best knock-down of *SNORD90* (*Figure 3—figure supplement 1K–L* & *Supplementary file 8*). NPCs were plated in 24-well plates until ~90% confluent before transfection with ASO (50 nM final concentration) using Lipofectamine 2000 (ThermoFisher) following manufacturer's protocol. NPCs were incubated for 48 hr before harvesting for RNA extraction. RNA extraction, cDNA synthesis, and RT-qPCR were as described above. One-way ANOVA was used as described above.

### RNA immunoprecipitation

Over-expression of *SNORD90* and scramble were performed as described above. RNA immunopre-cipitation (RIP) was performed using the Magna RIP RNA-Binding Protein Immunoprecipitation kit (Millipore Sigma) following the manufacturer's protocol with slight modifications. NPCs were collected and lysed in complete RIP lysis buffer. Ten uL of cell extract was stored at –80 °C and used as input. NPC extracts were incubated in RIP buffer containing magnetic beads conjugated to anti-fibrillarin antibody (Abcam, ab226178), anti-RBM15B antibody (Proteintech, 22249–1-AP), or IgG control (provided by the kit) overnight in 4 °C. Bead complexes were washed six times via genital inversion with RIP buffer and eluted directly in TRIzol. RNA from all conditions were purified in parallel using the RNA microPrep kit (Zymol) and eluted into 15 µL H2O. The entire eluate was transcribed to cDNA

using M-MLV Reverse Transcriptase (200 U/µL) (ThermoFisher) with random hexamers. RT-qPCR was as described above.

## m6A-RIP

m6A-RIP protocol was as described by Engel et al., with minor modifications. Three µg total RNA was mixed with 3 fmol spike-in and equally split into three conditions: m6A-RIP, IgG control, and input. Input samples were frozen and kept at –80 °C during the m6A-RIP protocol. m6A-RIP and IgG control samples were incubated with 1 µg anti-m6A antibody (Synaptic Systems, rabbit polyclonal 202 003) or 1 µg normal rabbit IgG (NEB) in immunoprecipitation (IP) buffer (10 mM Tris-HCl [pH 7.5], 150 mM NaCl, 0.1% IGEPAL CA-630 in nuclease-free H2O, 0.5 mL total volume) with 1 µL RNasin Plus (Promega) rotating head over tail at 4 °C for 2 hr, followed by incubation with 2 x washed 25 µL Dynabeads M-280 (Sheep anti-Rabbit IgG Thermo Fisher Scientific) rotating head over tail at 4 °C for 2 hr. Bead-bound antibody-RNA complexes were recovered on a magnetic stand and washed in the following order: twice with IP buffer, twice with high-salt buffer (10 mM Tris-HCl [pH 7.5], 500 mM NaCl, 0.1% IGEPAL CA-630 in nuclease-free H2O), and twice with IP buffer. RNA was eluted directly into TRIzol and input RNA was also taken up in TRIzol. RNA from all conditions was purified in parallel using the RNA microPrep (Zymol) and eluted into 15 µL H$_2$O. The entire eluate was transcribed to cDNA using M-MLV Reverse Transcriptase (200 U/µL) (Thermo Fisher) with random hexamers. RT-qPCR was as described above. One-way ANOVA was used as described above.

### Spike-in

A spike-in Oligo was used as a normalizer for quantitative RT-PCR. The spike-in oligo was 100 nt in length with 3 internal m6A/m sites within GGAC motif flanked by the most frequent nucleotides 5′ U/A, 3′ A/U, not complementary to hsa or mmu RefSeq mRNA or genome, secondary structure exposing m6A sites (as described by *Engel et al., 2018*). The sequence is:

GCAG AACC UAGU AGCG UGUGG mAC ACGA ACAG GUAU CAAU AUGC GGGU AUGG mACU AAAGCAACGUGCGAGAUUACGCUGAGG mACUACAAUCUCAGUUACCA (synthesized by Horizon Discoveries).

## Dicer-substrate short interfering RNA gene silencing in human NPC culture

Dicer-substrate short interfering RNA (dsiRNA) were initially validated via western blotting before the final experiment (*Figure 4—figure supplement 1C* & *Figure 5—figure supplement 1A–E*). Western blots were as described above with the following antibodies: YTHDF1 (Abcam, ab220162) at 1:5000, YTHDF2 (Abcam, ab220163) at 1:5000, YTHDF3 (Abcam, ab220163) at 1:5000, YTHDC1 (Abcam, ab122340) at 1:500, YTHDC2 (Abcam, ab220160) at 1:5000, RBM15B (Proteintech, 22249–1-AP) at 1:1000, and GAPDH (NEB, 3683 S) at 1:1000. Each knock-down was assessed again via RT-qPCR in the final experimental samples (*Figure 4—figure supplement 1D* & *Figure 5—figure supplement 1F–J*). NPCs were seeded in 24-well plates and grown to ~70% confluency. 50 nM dsiRNA was transfected using lipofectamine 2000 according to the manufacturer's instructions. A second transfection was performed 48 hr after the first transfection. Following the second round of dsiRNA transfection, *SNORD90* expression vectors were transfected 48 hrs later. NPCs were then collected 48 hr after the third transfection. DsiRNA sequences and oligos were provided by IDT (*Supplementary file 13*). RNA extraction, cDNA synthesis, and RT-qPCR were as described above. See *Supplementary file 12* for a list of primer sequences used. One-way ANOVA was used as described above.

## Electrophysiological recording

### Brain slices preparation

Mice were stereotactically injected with *Snord90* or Scramble viral vector as described above.

Three weeks after the injection, mice received an overdose injection of pentobarbital (100 mg/kg i.p.) and were perfused with carbogenated (95% O$_2$, 5% CO$_2$) ice-cold slicing solution containing (in mM): 2.5 KCl, 11 glucose, 234 sucrose, 26 NaHCO$_3$, 1.25 NaH$_2$PO$_4$, 10 MgSO$_4$, 2 CaCl$_2$; pH 7.4, 340 mOsm. After decapitation, 300 µm coronal slices containing the ACC were prepared in carbogenated ice-cold slicing solution using a vibratome (Leica VT 1200 S) and allowed to recover for 20 min at 33 °C in carbogenated high osmolarity artificial cerebrospinal fluid (high-Osm aCSF) containing (in mM): 3.2

KCl, 11.8 glucose, 132 NaCl, 27.9 NaHCO$_3$, 1.34 NaH$_2$PO$_4$, 1.07 MgCl$_2$, 2.14 CaCl$_2$; pH 7.4, 320 mOsm followed by 40 min incubation at 33 °C in carbogenated aCSF containing (in mM): 3 KCl, 11 glucose, 123 NaCl, 26 NaHCO$_3$, 1.25 NaH$_2$PO$_4$, 1 MgCl$_2$, 2 CaCl$_2$; pH 7.4, 300 mOsm. Subsequently, slices were kept at room temperature (RT) in carbogenated aCSF until use.

## Patch-clamp recordings

Slices were then transferred in the recording chamber and superfused (4–5 mL/min) with carbogenated aCSF and recordings performed at RT. Pyramidal neurons of the ACC were visualized with infrared differential interference contrast (DIC) microscopy (BX51W1, Olympus) and an Andor Neo sCMOS camera (Oxford Instruments, Abingdon, UK). Somatic whole-cell voltage-clamp recordings from ACC pyramidal neurons (>1 GΩ seal resistance, –70 mV holding potential) were performed using a Multiclamp 700B amplifier (Molecular Devices, San Jose, CA, USA). Data were acquired using pClamp 10.7 on a personal computer connected to the amplifier via a Digidata-1440 interface (sampling rate: 20 kHz; low-pass filter: 8 kHz). Borosilicate glass pipettes (BF100-58-10, Sutter Instrument, Novato, CA, USA) with resistances 4–6 MΩ were pulled using a laser micropipette puller (P-2000, Sutter Instrument). Data obtained with a series resistance >20 MΩ or fluctuation more than 20% of the initial values were discarded.

For sEPSCs recording, pyramidal neurons were clamped at –70 mV in the presence of BIM (20 uM) with the pipette solution containing (in mM): 125 Cs-methanesulfonate, 8 NaCl, 10 HEPES, 0.5 EGTA, 4 Mg-ATP, 0.3 Na-GTP, 20 Phosphocreatine and 5 QX-314 (pH 7.2 with CsOH, 285–290 mOsm).

Analysis was performed using ClampFit 10.7 and Easy Electrophysiology V2.2 (https://www.easy electrophysiology.com), and statistical significance assessed with GraphPad Prism 7.

# Acknowledgements

GT holds a Canada Research Chair (Tier 1) and is supported by grants from the Canadian Institute of Health Research (CIHR) (FDN148374, EGM141899, ENP161427), and by the Fonds de recherche du Québec -Santé (FRQS) through the Quebec Network on Suicide, Mood Disorders, and Related Disorders. CAN-BIND is an Integrated Discovery Program carried out in partnership with, and financial support from, the Ontario Brain Institute, an independent nonprofit corporation, funded partially by the Ontario government. The opinions, results, and conclusions are those of the authors and no endorsement by the Ontario Brain Institute is intended or should be inferred. Additional funding is provided by the Canadian Institutes of Health Research (CIHR). All study medications were independently purchased at wholesale market values. AC is the incumbent of the Vera and John Schwartz Family Professorial Chair in Neurobiology at the Weizmann Institute and the head of the Max Planck Society–Weizmann Institute of Science Laboratory for Experimental Neuropsychiatry and Behavioral Neurogenetics. This work is supported by the German Ministry of Science and Education (IMADAPT, FKZ: 01KU1901); the Ruhman Family Laboratory for Research in the Neurobiology of Stress (AC); research support from Bruno and Simone Licht; the Perlman Family Foundation, founded by Louis L and Anita M Perlman (AC); the Adelis Foundation (AC); and Sonia T Marschak (AC). JPL holds postdoctoral fellowships from the European Molecular Biology Organization (EMBO-ALTF 650–2016), Alexander von Humboldt Foundation, and the Canadian Biomarker Integration Network in Depression (CAN-BIND). JD is the incumbent of the Achar Research Fellow Chair in Electrophysiology. Images created using https://www.biorender.com/.

# Additional information

## Competing interests

Raymond W Lam: RM has received consulting and speaking honoraria from AbbVie, Allergan, Eisai, Janssen, KYE, Lallemand, Lundbeck, Neomind, Otsuka, and Sunovion, and research grants from CAN-BIND, CIHR, Janssen, Lallemand, Lundbeck, Nubiyota, OBI and OMHF. Jane A Foster: JAF has received consulting and speaking fees from Takeda and RBH, and research funding from NSERC, CIHR, and OBI. The other authors declare that no competing interests exist.

## Funding

| Funder | Grant reference number | Author |
| --- | --- | --- |
| Canadian Institutes of Health Research | FDN148374 | Gustavo Turecki |
| Canadian Institutes of Health Research | EGM141899 | Gustavo Turecki |
| Canadian Institutes of Health Research | ENP161427 | Gustavo Turecki |
| Bundesministerium für Bildung und Forschung | IMADAPT | Alon Chen |
| Bundesministerium für Bildung und Forschung | FKZ:01KU1901 | Alon Chen |
| Ruhman Family Laboratory for Research in the Neurobiology of Stress | | Alon Chen |
| Bruno and Simone Licht | | Alon Chen |
| Perlman Family Foundation | | Alon Chen |
| Adelis Foundation | | Alon Chen |
| Sonia T Marschak | | Alon Chen |
| European Molecular Biology Organization | Fellowship EMBO-ALTF 650–2016 | Juan Pablo Lopez |
| Alexander von Humboldt Foundation | Fellowship | Juan Pablo Lopez |
| Canadian Biomarker Integration Network in Depression | Fellowship | Juan Pablo Lopez |
| Achar Research Fellow Chair in Electrophysiology | | Julien Dine |

The funders had no role in study design, data collection and interpretation, or the decision to submit the work for publication.

## Author contributions

Rixing Lin, Conceptualization, Data curation, Formal analysis, Validation, Investigation, Methodology, Writing - original draft, Writing – review and editing; Aron Kos, Data curation, Formal analysis, Investigation, Visualization, Methodology, Writing – review and editing; Juan Pablo Lopez, Conceptualization, Data curation, Methodology, Writing – review and editing; Julien Dine, Data curation, Investigation, Visualization, Methodology, Writing – review and editing; Laura M Fiori, Data curation, Investigation, Methodology, Writing – review and editing; Jennie Yang, Yair Ben-Efraim, Pascal Ibrahim, Haruka Mitsuhashi, Data curation; Zahia Aouabed, Software, Formal analysis; Tak Pan Wong, Methodology; El Cherif Ibrahim, Catherine Belzung, Pierre Blier, Faranak Farzan, Benicio N Frey, Raymond W Lam, Roumen Milev, Daniel J Muller, Sagar V Parikh, Claudio Soares, Rudolf Uher, Corina Nagy, Naguib Mechawar, Jane A Foster, Sidney H Kennedy, Data curation, Writing – review and editing; Alon Chen, Resources, Funding acquisition, Investigation, Methodology; Gustavo Turecki, Conceptualization, Resources, Supervision, Funding acquisition, Investigation, Methodology, Writing - original draft, Project administration, Writing – review and editing

## Author ORCIDs

Tak Pan Wong  http://orcid.org/0000-0001-8611-4911
El Cherif Ibrahim  http://orcid.org/0000-0003-3973-7862
Raymond W Lam  http://orcid.org/0000-0001-7142-4669
Corina Nagy  http://orcid.org/0000-0003-1439-0129
Alon Chen  http://orcid.org/0000-0003-3625-8233
Gustavo Turecki  http://orcid.org/0000-0003-4075-2736

## Ethics

Clinical trial registration see methods.

Human subjects: Clinical Cohort 1: This clinical trial was approved by ethics boards of participating centers, and all participants provided written informed consent. https://www.clinicaltrials.gov/ (11984A NCT00635219; 11918A NCT00599911; 13267A NCT01140906).Clinical Cohort 2: This trial was approved by ethics boards of participating centers and all participants provided written informed consent. https://www.clinicaltrials.gov/ identifier NCT01655706.Clinical Cohort 3: All subjects included in the study provided informed consent, and the project was approved by The Institutional Review Board of the Douglas Mental Health University Institute.Post-mortem Cohort: Written informed consent was obtained from next-of-kin. This study was approved by the Douglas Hospital Research Centre institutional review board.

All experiments on mice were carried out according to policies on the care and use of laboratory animals of European Community legislation 2010/63/EU. All protocols were approved by the local Ethics Committee (CEEAVdL-19; protocol number 2011-06-10), the Ethics Committee for the Care and Use of Laboratory Animals of the government of Upper Bavaria, Germany, and by the Institutional Animal Care and Use Committee (IACUC) of the Weizmann Institute of Science (Rehovot, Israel).

## Decision letter and Author response

Decision letter https://doi.org/10.7554/eLife.85316.sa1
Author response https://doi.org/10.7554/eLife.85316.sa2

# Additional files

## Supplementary files

• Supplementary file 1. Summary statistics for small RNA sequencing analysis in human clinical discovery cohort.

• Supplementary file 2. Summary statistics for small RNA sequencing analysis in human clinical replication 1 cohort.

• Supplementary file 3. Summary statistics for small RNA sequencing analysis in human clinical replication 2 cohort.

• Supplementary file 4. Sequences for over-expression.

• Supplementary file 5. *SNORD90* BLAST target prediction. Genes highlighted in green were selected for wet lab confirmation.

• Supplementary file 6. *SNORD90* PLEXY target prediction (whole genome). Transcripts highlighted in green were selected for wet lab confirmation. Target sites in bold are sites used for target blocker design. (a) Parenthesis indicates complimentary base match between *SNORD90* and target site. Dot indicates mismatch. (b) Sequence of the target site and site on *SNORD90*. Match column indicates sequence complementarity or mismatch shown here. m: 2'-O-ribose methylation site.

• Supplementary file 7. *SNORD90* PLEXY target prediction (*NRG3* targeted). Target sites in bold are sites used for target blocker design. (a) Parenthesis indicates complimentary base match between *SNORD90* and target site. Dot indicates mismatch. (b) Sequence of the target site and site on *SNORD90*. Match column indicates sequence complementarity or mismatch shown here. m: 2'-O-ribose methylation site.

• Supplementary file 8. ASO sequences. m: 2'-O-methyl base; *: Phosphorothioate bond

• Supplementary file 9. Target blocker sequences. 2MOEr: 2'-O-methoxyethyl RNA base; /*/: Phosphorothioate bond

• Supplementary file 10. RBP motif on *SNORD90*. (a) A value of 0 and 1 is assigned for each putative RBP motif identified on SNORD90 where 1 is the canonical motif. (b) Corresponds to the probability of the 7-mer region being structurally unpaired (lower score corresponds to lower probability)

• Supplementary file 11. *Snord90* PLEXY target prediction (whole genome). Target sites highlighted in green display the highest probability for *Snord90* interaction on *Nrg3* (a) Parenthesis indicates complimentary base match between *SNORD90* and target site. Dot indicates mismatch.
(b) Sequence of the target site and site on *SNORD90*. Match column indicates sequence complementarity or mismatch shown here. m: 2'-O-ribose methylation site.

• Supplementary file 12. Primer sequences.

- Supplementary file 13. DsiRNA sequences.
- MDAR checklist

## Data availability

Gene expression data have been deposited in the GEO database under the accession code GSE224050.

The following dataset was generated:

| Author(s) | Year | Dataset title | Dataset URL | Database and Identifier |
| --- | --- | --- | --- | --- |
| Turechi G | 2023 | SNORD90 induces glutamatergic signaling following treatment with monoaminergic antidepressants | http://www.ncbi.nlm.nih.gov/geo/query/acc.cgi?acc=GSE224050 | NCBI Gene Expression Omnibus, GSE224050 |

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
