## [Editor Report]

This is an important study that uncovers a new molecular pathway that links traditional monoaminergic antidepressants with regulation of glutamate neurotransmission. The data provided for the model are convincing and demonstrate the pathway in human plasma and brain, mouse brain, and cultured cells, using the relative strengths of each system. The work will be of interest to psychiatrists studying depression as well as basic neurobiologists interested in monoamine signaling in the brain.

---

## [Decision Letter]

**Decision letter after peer review:**

Thank you for submitting your article "SNORD90 induces glutamatergic signaling following treatment with monoaminergic antidepressants" for consideration by *eLife*. Your article has been reviewed by 2 peer reviewers, one of whom is a member of our Board of Reviewing Editors, and the evaluation has been overseen by Ma-Li Wong as the Senior Editor. The reviewers have opted to remain anonymous.

Essential revisions:

1) Both reviewers point out that it is hard to interpret the study without knowing which cell types express SNORD90. Ideally the authors would provide experimental data to address this important point.

2) Although the reviewers felt that the molecular pathway was generally well supported by the data, both expressed some concerns about the relationship and relative importance of the molecular data to the context of antidepressant action. The authors should build out a much more thorough introduction and discussion for the study that places this pathway in context and offers alternative interpretations or future experiments that can expand on the data offered here. The reviewers suggest some possible experiments that could tighten these links, but no additional experiments are required to resolve this concern – discussion is sufficient.

3) Please make sure to address the concerns the reviewers raised about key details missing from the methods or figures.

*Reviewer #1 (Recommendations for the authors):*

The story begins with the authors screening peripheral blood from patients in an antidepressant clinical trial for small RNAs, including snoRNAs. The identification of SNORD90 as differential in plasma held across three independent replicate cohorts treated with different drugs. The in vivo changes in ACC of humans are less robust, as might be expected, but they also find antidepressant inducible Snord90 expression in mouse ACC and cultured human neurons.

– The authors presumably saw a wide variety of small RNAs in their sequencing, so a sentence saying why they focus on the snoRNAs here would be helpful. The introduction is concise, so including a paragraph about the utility and/or challenges of measuring small RNAs in plasma and the precedence for thinking they are important would ground the study.

– Why did the authors use monoaminergic neurons in Figure 1? If the authors think forebrain neurons in the ACC that receive monoaminergic signaling are the ones regulated by SNORD90, why didn't they differentiate glutamatergic neurons? Or is SNORD90 expressed in most (or all) neurons or cell types? This figure would suggest the expression is more broad than narrow.

– What cell type do the authors think is responsible for the SNORD90 RNA recovered in plasma? This again raises the question – is Snord90 expressed everywhere? Can the authors show a cell type where antidepressants do not regulate Snord90 expression?

The authors then go on to overexpress Snord90 in the cingulate cortex of the mouse brain and study behaviors. Overexpression, in this case, makes sense given that the authors find that antidepressants raise Snord90 levels.

– It would help ground the model to know which cells in the cingulate show upregulation of Snord90 in response to antidepressants and to validate that the overexpression enhances expression in the same cell types.

The authors next identify NRG3 as a Snord90 target based on sequence complementarity. Although, this section is thorough, the binding site scrambled control in Figure 3 is strong, as was the effort to knockdown each of the YTH proteins.

– Focusing on NRG3 is acceptable for the story, but it would be helpful to see a list of all possible targets that were rejected for study before selecting this one.

The last part of the study shows evidence that glutamatergic neurotransmission is altered when Snord90 is overexpressed. However, evidence of this via RNA methylation, NRG expression, and SNARE disruption is somewhat tenuous. The paper cited shows that Nrg3 can disrupt SNAREs, but there is no evidence that it does that in these cells under these conditions.

– The key experiment would be to knockout NRG3 in ACC of mice with Snord90 overexpression and determine if that blocks the antidepressant behavior, parallel to the study in Figure 2. If the authors did this, it would substantially increase the connection of the molecular pathway to the behavior. However, it is reasonable if the authors consider this to be beyond the scope of the current study. What is required is that the authors discuss possible alternative interpretations of this study. The Discussion section needs to be revised for the data presented and should be expanded to contextualize the findings in the current literature and talk about possible future studies that could test the importance of the mechanisms outlined here.

*Reviewer #2 (Recommendations for the authors):*

Whereas the manuscript is well written overall, the rationale for studying the patterns of snoRNA expression following antidepressant treatment is not clearly defined in the manuscript. Addressing this could improve the flow at the beginning of the Results section.

Whereas many putative targets of SNORD90 are described, validation of several targets, not just NRG3, would demonstrate the strength of the in silico analysis conducted here.

The effects of antidepressants are thought to be through the regulation of glutamatergic neurotransmission, but SNORD90 levels seem to be upregulated across various cell types and tissues (peripheral blood and brain tissue) following antidepressant treatment. NRG3 downregulation by SNORD90 is also non-specific and occurs within neural progenitors. Thus, the significance of the SNORD90-NRG3 nexus in mediating specific behavioral and excitability changes needs to be clarified. For instance, could the reduced binding of SNORD90 specifically to NRG3 mRNA suppress the increase in glutamatergic neurotransmission?

The Materials and methods section and the text need more detail in describing the in silico target prediction efforts. However, the presentation of the results seems to suggest that SNORD90 has a single target in NRG3 and recruits a single RNA reader protein that functions non-canonically to regulate NRG3 expression. Precisely how the information presented in supplementary tables 4 and 5 to select NRG3 should be clarified.

A major unanswered question in the manuscript is that, whereas SNORD90 binds to intronic sequences, methylation of mature NRG3 mRNA is also elevated by SNORD90 overexpression. Is methylation of NRG3 pre-mRNA restricted to the vicinity of SNORD90 binding, or are pre-mRNAs also methylated in exons? Addressing this question could provide greater clarity into the proposed mechanism of action of SNORD90 and validate its significance.

Addressing these concerns should render the manuscript suitable for publication.

---

## [Author Response]

Essential revisions:1) Both reviewers point out that it is hard to interpret the study without knowing which cell types express SNORD90. Ideally the authors would provide experimental data to address this important point.

Thank you for this suggestion. We have collected new data using fluorescence-activated nuclei sorting (FANS) whereby we sorted nuclei collected from our human port-mortem ACC samples into NeuN+ (neuronal) or NeuN- (non-neuronal) cell types. This new data suggests that the up-regulation of *SNORD90* in response to antidepressants is not occurring in a cell type specific manner. Accordingly, the group of individuals with MDD with antidepressants is significantly up-regulated in both neuronal and non-neuronal fractions, even though there are some differences in post-hoc comparisons between subgroups.

This has been integrated into the Results section and Figure 1—figure supplement 1B as indicated below (also see revised manuscript lines 120-128).

“To further asses if there is any cell type specificity to the effects of antidepressants to the up-regulation of SNORD90 we employed fluorescence-activated nuclei sorting (FANS), using samples from the human post-mortem ACC tissue described above, to separate neuronal (NeuN+ nuclei) and non-neuronal (NeuN- nuclei) and measured the expression of SNORD90 in each nuclei fraction. In both the neuronal and non-neuronal nuclei fractions we observed the same pattern of expression as in bulk tissue, where SNORD90 was up-regulated in individuals with depression that were actively treated with antidepressants at the time of death (Figure 1—figure supplement 1B). These results suggest that antidepressants do not up-regulate SNORD90 in a cell-type specific manner.”

2) Although the reviewers felt that the molecular pathway was generally well supported by the data, both expressed some concerns about the relationship and relative importance of the molecular data to the context of antidepressant action. The authors should build out a much more thorough introduction and discussion for the study that places this pathway in context and offers alternative interpretations or future experiments that can expand on the data offered here. The reviewers suggest some possible experiments that could tighten these links, but no additional experiments are required to resolve this concern – discussion is sufficient.

Thank you for this suggestion. We have updated our introduction and Discussion sections to address the comments and questions from the reviewers as indicated below (also see revised manuscript lines 84-92 and lines 336-424).

Addition to introduction (revised manuscript lines 84-92):

“The functional activity of genes is at the core of all biological processes. Thus, investigating the molecular factors that modulate gene expression in relation to antidepressant treatment should provide better insight into the molecular mechanisms surrounding antidepressant response. Non-coding RNAs act as fine tuners of gene expression through a diverse range of functional mechanisms (10). In this study, we focused our attention on a class of small non-coding RNA called small nucleolar RNAs (snoRNAs). Although snoRNAs have classically been associated with housekeeping roles they have more recently been shown to be involved in complex regulatory roles in gene expression such as regulation of alternative splicing, precursor to smaller miRNA-like RNA fragments, and direct regulation of mRNA expression (11-13).”

Revised Discussion (revised manuscript lines 336-424):

“In this study, we investigated the expression of snoRNAs in peripheral blood collected across three independent clinical cohorts from patients with MDD that underwent eight weeks of antidepressant treatment. From this investigation our results pointed to an up-regulation of SNORD90 having a consistent association with antidepressant treatment response across all three of our clinical cohorts. Although other snoRNAs were also identified to have an association with antidepressant response, they were not consistently differentially expressed across all three of our clinical cohorts. Through a series of experiments using rodent models, post-mortem human brain samples, and human neuronal cultures we further determined that the up-regulation of SNORD90 is primarily achieved by antidepressant drugs and not by other psychotropic drugs. It is unclear why SNORD90 seems to be preferentially up-regulated by antidepressant drugs, but we speculate it is due to the mode of action of antidepressant drugs, many of which modulate the serotonergic system. Serotonin signaling via binding to serotonin receptors has been shown to be linked with chromatin remodeling thereby modulating gene expression (34). Accordingly, a recent study points to the ability of serotonin to modify histones. Specifically, the glutamine at the Q5 site in histone H3 can be modified by serotonin which promotes the recruitment of transcription factor II D (TFIID) and promotes gene expression (35). This process, however, is dependent on intracellular levels of serotonin, more specifically, the levels of serotonin in the nucleus, and it remains unclear if antidepressants can induce such changes.

[…]

In addition to the contribution to understanding antidepressant drug action we also contribute to the growing knowledge pertaining to the functional mechanisms of snoRNAs. Beyond their canonical regulatory roles, snoRNAs have been reported to influence gene expression via interaction with RNA targets however the mechanism by which this occurs remained unclear. We are the first to propose that snoRNAs play a role in m6A modifications which ultimately has regulatory roles in the stability of mRNA.”

3) Please make sure to address the concerns the reviewers raised about key details missing from the methods or figures.

More details have been added to the methods related to the in-silico target prediction as requested by the reviewers as indicated below (also see revised manuscript lines 817-832).

“The entire sequence of SNORD90 was used to blast against the human genome allowing G-T wobble base pairing and one mismatch base pairing (Supplementary File 5). From this prediction we identified 100 locations on the human genome (both within genes and gene desert regions) that has complementarity sequences to SNORD90. NRG3 was ranked as one of highest genes housing a complementarity sequence to SNORD90 without any mismatches. To increase our confidence in target selection we also utilized the C/D box snoRNA target prediction algorithm PLEXY, using default parameters (18). SNORD90 mature sequence was used as the input snoRNA and target sequence input was the entire human genome downloaded from the Ensembl genome browser database (Supplementary File 6-7). Prediction was conducted using whole transcriptomics (Supplementary File 6) as well as a targeted analysis for NRG3 (Supplementary File 7). Using the PLEXY algorithm also identified NRG3 as a possible target for SNORD90 (Supplementary File 6). Although NRG3 was not ranked among the highest predicted targets using the PLEXY algorithm, NRG3 was selected as the candidate gene target since it was the only gene that displayed overlap between the two target prediction methods (Supplementary File 5-6). PLEXY further reviled that SNORD90 has multiple possible interaction sites on NRG3, thus we selected the top three predicted sites for consideration in future experiments (figure 3A & Supplementary File 6-7).”

All figures and figure legends have been updated to make clearer the organism of origin for each experiment (please see all revised figures and figure legends).

Reviewer #1 (Recommendations for the authors):The story begins with the authors screening peripheral blood from patients in an antidepressant clinical trial for small RNAs, including snoRNAs. The identification of SNORD90 as differential in plasma held across three independent replicate cohorts treated with different drugs. The in vivo changes in ACC of humans are less robust, as might be expected, but they also find antidepressant inducible Snord90 expression in mouse ACC and cultured human neurons.

– The authors presumably saw a wide variety of small RNAs in their sequencing, so a sentence saying why they focus on the snoRNAs here would be helpful. The introduction is concise, so including a paragraph about the utility and/or challenges of measuring small RNAs in plasma and the precedence for thinking they are important would ground the study.

Thank you for this suggestion and we added more details to the revised introduction as indicated below (also see revised manuscript lines 84-92).

“The functional activity of genes is at the core of all biological processes. Thus, investigating the molecular factors that modulate gene expression in relation to antidepressant treatment should provide better insight into the molecular mechanisms surrounding antidepressant response. Non-coding RNAs act as fine tuners of gene expression through a diverse range of functional mechanisms (10). In this study, we focused our attention on a class of small non-coding RNA called small nucleolar RNAs (snoRNAs). Although snoRNAs have classically been associated with housekeeping roles they have more recently been shown to be involved in complex regulatory roles in gene expression such as regulation of alternative splicing, precursor to smaller miRNA-like RNA fragments, and direct regulation of mRNA expression (11-13).”

– Why did the authors use monoaminergic neurons in Figure 1? If the authors think forebrain neurons in the ACC that receive monoaminergic signaling are the ones regulated by SNORD90, why didn't they differentiate glutamatergic neurons? Or is SNORD90 expressed in most (or all) neurons or cell types? This figure would suggest the expression is more broad than narrow.

We used monoaminergic neurons since we were investigating monoaminergic antidepressants. At the time of this particular experiment and at this stage of the project we had no insight into the possible implications this would have to the glutamatergic system. This explains why we did not investigate glutamatergic neurons.

Yes, SNORD90 seems to be expressed in many if not all tissue types and in most or likely all cell types (Fafard-Couture et al., 2021; PMID: 34088344). The reviewer is correct in saying that the expression of SNORD90 is broad and not narrow. We ultimately narrowed down to the glutamatergic neurons not because of SNORD90’s pattern of expression but SNORD90’s ability to regulate NRG3, which is primarily expressed in glutamatergic neurons.

– What cell type do the authors think is responsible for the SNORD90 RNA recovered in plasma? This again raises the question – is Snord90 expressed everywhere? Can the authors show a cell type where antidepressants do not regulate Snord90 expression?

We extracted total RNA from white blood cells (leukocytes and lymphocytes). It is plausible to hypothesize that treatment with antidepressants will alter the expression of SNORD90 regardless of cell type.

The authors then go on to overexpress Snord90 in the cingulate cortex of the mouse brain and study behaviors. Overexpression, in this case, makes sense given that the authors find that antidepressants raise Snord90 levels.– It would help ground the model to know which cells in the cingulate show upregulation of Snord90 in response to antidepressants and to validate that the overexpression enhances expression in the same cell types.The authors next identify NRG3 as a Snord90 target based on sequence complementarity. Although, this section is thorough, the binding site scrambled control in Figure 3 is strong, as was the effort to knockdown each of the YTH proteins.– Focusing on NRG3 is acceptable for the story, but it would be helpful to see a list of all possible targets that were rejected for study before selecting this one.

We did not use cell type specific promoters in our over-expression plasmids, thus the over-expression of Snord90 is occurring in all cells that take-up the plasmid (identified via eGFP expression).

We have collected new data using fluorescence-activated nuclei sorting (FANS) whereby we sorted nuclei collected from our human port-mortem ACC samples into NeuN+ (neuronal) or NeuN- (non-neuronal) cell types. This new data suggests that the up-regulation of *SNORD90* in response to antidepressants is not occurring in a cell type specific manner. Accordingly, the group of individuals with MDD with antidepressants is significantly up-regulated in both neuronal and non-neuronal fractions, even though there are some differences in post-hoc comparisons between subgroups.

This has been integrated into the Results section and Figure 1—figure supplement 1B as indicated below (also see revised manuscript lines 120-128).

“To further asses if there is any cell type specificity to the effects of antidepressants to the up-regulation of SNORD90 we employed fluorescence-activated nuclei sorting (FANS), using samples from the human post-mortem ACC tissue described above, to separate neuronal (NeuN+ nuclei) and non-neuronal (NeuN- nuclei) and measured the expression of SNORD90 in each nuclei fraction. In both the neuronal and non-neuronal nuclei fractions we observed the same pattern of expression as in bulk tissue, where SNORD90 was up-regulated in individuals with depression that were actively treated with antidepressants at the time of death (Figure 1—figure supplement 1B). These results suggest that antidepressants do not up-regulate SNORD90 in a cell-type specific manner.”

We have selected six other predicted targets to validate in our in-vitro SNORD90 over-expression and SNORD90 knock-down experiments in human NPC culture. From these data we also identified another target, ENG, that was up-regulated with SNORD90 over-expression and down-regulated with SNORD90 knock-down.

ENG is a type I transmembrane glycoprotein, part of the transforming growth factor β receptor complex and is primarily expressed with activated endothelial cells playing a major role in angiogenesis both in development and tumor progression [Schoonderwoerd et al., 2020; PMID: 32059544]. Since NRG3 is primarily expressed in the central nervous system and has been linked to psychiatric disorders in previous studies, we focused our attention exclusively on NRG3 in downstream experiments.

This has been integrated into the Results section of the revised manuscript (lines 184-191 & 201-204 & 209-220) and Figure 1—figure supplement 1E-J and 1M-R.

The last part of the study shows evidence that glutamatergic neurotransmission is altered when Snord90 is overexpressed. However, evidence of this via RNA methylation, NRG expression, and SNARE disruption is somewhat tenuous. The paper cited shows that Nrg3 can disrupt SNAREs, but there is no evidence that it does that in these cells under these conditions.– The key experiment would be to knockout NRG3 in ACC of mice with Snord90 overexpression and determine if that blocks the antidepressant behavior, parallel to the study in Figure 2. If the authors did this, it would substantially increase the connection of the molecular pathway to the behavior. However, it is reasonable if the authors consider this to be beyond the scope of the current study.

We thank the reviewer for these comments and interesting suggestions for additional experiments. We will plan to do this work in future studies. We have included a sentence in this regard in the revised discussion which reads “Future studies concomitantly manipulating levels of Nrg3 and Snord90 will be important to further test the effects noted in this study.” (also see revised manuscript lines 391-392)

What is required is that the authors discuss possible alternative interpretations of this study. The Discussion section needs to be revised for the data presented and should be expanded to contextualize the findings in the current literature and talk about possible future studies that could test the importance of the mechanisms outlined here.

We thank the Reviewer for these suggestions. The discussion has been revised as indicated below (also see revised manuscript lines 336-424).

“In this study, we investigated the expression of snoRNAs in peripheral blood collected across three independent clinical cohorts from patients with MDD that underwent eight weeks of antidepressant treatment. From this investigation our results pointed to an up-regulation of SNORD90 having a consistent association with antidepressant treatment response across all three of our clinical cohorts. Although other snoRNAs were also identified to have an association with antidepressant response, they were not consistently differentially expressed across all three of our clinical cohorts. Through a series of experiments using rodent models, post-mortem human brain samples, and human neuronal cultures we further determined that the up-regulation of SNORD90 is primarily achieved by antidepressant drugs and not by other psychotropic drugs. It is unclear why SNORD90 seems to be preferentially up-regulated by antidepressant drugs, but we speculate it is due to the mode of action of antidepressant drugs, many of which modulate the serotonergic system. Serotonin signaling via binding to serotonin receptors has been shown to be linked with chromatin remodeling thereby modulating gene expression (34). Accordingly, a recent study points to the ability of serotonin to modify histones. Specifically, the glutamine at the Q5 site in histone H3 can be modified by serotonin which promotes the recruitment of transcription factor II D (TFIID) and promotes gene expression (35). This process, however, is dependent on intracellular levels of serotonin, more specifically, the levels of serotonin in the nucleus, and it remains unclear if antidepressants can induce such changes.

A combination of in-silico target prediction and a series of gain and loss of SNORD90 function assays indicates that SNORD90 is able to modulate the expression of mRNA transcripts, particularly NRG3. Using additional in-silico prediction tools and follow-up wet lab experiments we identified a high confidence sequence motif on SNORD90 for RBM15B, an essential component of the larger m6A methyltransferase complex (25, 26). This is a significant divergence from canonically functioning C/D box snoRNAs that are known to associate with fibrillarin, a methyltransferase responsible for 2’-O-methylation (2’OMe) (36, 37). Whereas 2’OMe can occur on the ribose of any base and is associated with increased RNA stability, m6A is an adenosine specific modification with a diverse range of effects on RNA stability (33, 38). We followed up by measuring m6A abundance on pre-NRG3 and NRG3 and found that elevated levels of SNORD90 is associated with increased m6A abundance on both pre-NRG3 and NRG3. Together our results suggest that SNORD90 is acting as a guide RNA to sequester the m6A-writer complex onto target transcripts in the nucleus. We further determined that the increase of m6A abundance on NRG3, as a result of up-regulated SNORD90 expression, mediates NRG3 mRNA decay via recognition from the m6A-reader YTHDF2. NRG3 is a member of the neuregulin family of epidermal growth factor like signaling molecules involved in cell-to-cell communication (39). There are four known human neuregulin paralogs: NRG1, NRG2, NRG3, and NRG4. NRG1 is the most well studied, however there has been growing interest in NRG3 in neuropsychiatric research. NRG3 was first identified in 1997 and described as an erb-b2 receptor tyrosine kinase 4 (ERBB4) ligand enriched in neural tissue (21). Additionally, according to the Genotype-Tissue Expression (GTEx) project, NRG3 is almost exclusively expressed in the central nervous system with the highest expression in the anterior cingulate and frontal cortex; NRG3 also displays the strongest specificity to the central nervous system as compared to the other neuregulin paralogs (39). NRG3 has been associated with several psychiatric illnesses, primarily schizophrenia but also MDD and bipolar disorder; however, the functional relevance of NRG3 remains poorly understood (19, 40-42). A more recent study suggested that NRG3 acts in a cell-autonomous manner specifically in pyramidal neurons and not by activating ERBB4 receptors (20). They propose that NRG3 interacts with syntaxin 1(specially at the SNARE domain), at the pre-synaptic terminal, resulting in the inhibition of SNARE-complex formation preventing vesicle docking and reducing glutamate release (20). Furthermore, by utilizing a NRG3 knock-out mouse model, they observed a significant increase in glutamatergic signaling (20). This finding was of significant interest to us since our data points to a down-regulation of NRG3 mediated by up-regulation of SNORD90 after antidepressant treatment. To determine if our SNORD90-NRG3 regulatory network has implications on glutamatergic signaling we over-expressed SNORD90 in the mouse anterior cingulate cortex followed by slice electrophysiology recordings where we observed an increase in spontaneous excitatory post synaptic currents (sEPSC). Future studies concomitantly manipulating levels of Nrg3 and Snord90 will be important to further test the effects noted in this study.

In summary our proposed model suggests that antidepressant treatment is linked to an up-regulation of SNORD90, which in turn recruits the m6A-writer complex and guides this complex onto pre-NRG3 increasing the abundance of m6A modifications which is further retained onto the mature NRG3 transcript. In turn, the greater abundance of m6A modifications on NRG3 is recognized by the m6A-reader, YTHDF2, which mediates NRG3 decay. This reduction in NRG3 levels is associated with an increase in glutamatergic signaling which, we believe, contributes to antidepressant response.

The glutamatergic system has become of great interest in antidepressant therapeutics research since drugs targeting this system, in particular ketamine, results in rapid alleviation of MDD symptoms. It has been repeatedly shown that monoaminergic antidepressants have associative effects on the glutamatergic system, however the exact molecular mechanistic link between the two system has remained unknown (9). Our findings suggest that snoRNAs play a role in antidepressant treatment response and provides a molecular link between monoaminergic targeting antidepressants to implications to the glutamatergic system. Several theories postulate that the therapeutic effects of ketamine and other rapid acting antidepressant is through NMDA receptor inhibition. Among these theories, the disinhibition hypothesis suggests that ketamine may be preferentially inhibiting NMDA receptors located on GABAergic interneurons thus disinhibiting excitatory pyramidal neurons leading to an enhancement of glutamatergic neurotransmission. In contrast other theories propose that ketamine directly inhibits glutamatergic neurotransmission via NMDAR inhibition (43). Our findings support the notion that the monoamine system may be playing more of a modularly role to the glutamatergic system to achieve an antidepressant outcome; in particular we observed that an increase in glutamatergic neurotransmission is associated with a response to monoaminergic-acting antidepressant.

In addition to the contribution to understanding antidepressant drug action we also contribute to the growing knowledge pertaining to the functional mechanisms of snoRNAs. Beyond their canonical regulatory roles, snoRNAs have been reported to influence gene expression via interaction with RNA targets however the mechanism by which this occurs remained unclear. We are the first to propose that snoRNAs play a role in m6A modifications which ultimately has regulatory roles in the stability of mRNA.”

Reviewer #2 (Recommendations for the authors):Whereas the manuscript is well written overall, the rationale for studying the patterns of snoRNA expression following antidepressant treatment is not clearly defined in the manuscript. Addressing this could improve the flow at the beginning of the Results section.

We thank the Reviewer for this suggestion. We have added more details in the introduction as follows (see revised manuscript lines 84-92):

“The functional activity of genes is at the core of all biological processes. Thus, investigating the molecular factors that modulate gene expression in relation to antidepressant treatment should provide better insight into the molecular mechanisms surrounding antidepressant response. Non-coding RNAs act as fine tuners of gene expression through a diverse range of functional mechanisms (10). In this study, we focused our attention on a class of small non-coding RNA called small nucleolar RNAs (snoRNAs). Although snoRNAs have classically been associated with housekeeping roles they have more recently been shown to be involved in complex regulatory roles in gene expression such as regulation of alternative splicing, precursor to smaller miRNA-like RNA fragments, and direct regulation of mRNA expression (11-13).”

Whereas many putative targets of SNORD90 are described, validation of several targets, not just NRG3, would demonstrate the strength of the in silico analysis conducted here.

We have selected six other predicted targets to validate (three from each of our in-silico prediction approaches) in our in-vitro SNORD90 over-expression and SNORD90 knock-down experiments in human NPC culture. From these data we also identified another target, ENG, that was up-regulated with SNORD90 over-expression and down-regulated with SNORD90 knock-down.

ENG is a type I transmembrane glycoprotein, part of the transforming growth factor β receptor complex and is primarily expressed with activated endothelial cells playing a major role in angiogenesis both in development and tumor progression [Schoonderwoerd et al., 2020; PMID: 32059544]. Since NRG3 is primarily expressed in the central nervous system and has been linked to psychiatric disorders in previous studies, we focused our attention exclusively on NRG3 in downstream experiments.

This has been integrated into the Results section of the revised manuscript (lines 184-191 &201-204 & 209-220) and Figure 1—figure supplement 1E-J and 1M-R.

The effects of antidepressants are thought to be through the regulation of glutamatergic neurotransmission, but SNORD90 levels seem to be upregulated across various cell types and tissues (peripheral blood and brain tissue) following antidepressant treatment. NRG3 downregulation by SNORD90 is also non-specific and occurs within neural progenitors. Thus, the significance of the SNORD90-NRG3 nexus in mediating specific behavioral and excitability changes needs to be clarified. For instance, could the reduced binding of SNORD90 specifically to NRG3 mRNA suppress the increase in glutamatergic neurotransmission?

The ability for SNORD90 to regulate its targets does not seem to be cell type specific, however the enrichment of its targets does display a degree of cell/tissue type specificity. NRG3 is primarily expressed in excitatory neurons and associates with glutamatergic signaling. However, it is certainly possible that the downregulatory effects of NRG3 in a different neuronal cell type may have different implications.

The Materials and methods section and the text need more detail in describing the in silico target prediction efforts. However, the presentation of the results seems to suggest that SNORD90 has a single target in NRG3 and recruits a single RNA reader protein that functions non-canonically to regulate NRG3 expression. Precisely how the information presented in supplementary tables 4 and 5 to select NRG3 should be clarified.

Thank you for pointing this out.

The following has been added to Results section to clarify that NRG3 was not the only predicted target but was selected because it was the only gene to show overlap between the two in-silico methods (also see revised manuscript lines 169-172):

“Using these in-silico methods wedicted targets for SNORD90 and selected neuregulin 3 (NRG3) as a putative gene target because it was the only candidate gene to be predicted by both in-silico approaches (Supplementary File 5-6; see methods for more details).”

Additional descriptions have also been added to the methods section, which now reads as the following (also see revised manuscript lines 816-832):

“The entire sequence of SNORD90 was used to blast against the human genome allowing G-T wobble base pairing and one mismatch base pairing (Supplementary File 5). From this prediction we identified 100 locations on the human genome (both within genes and gene desert regions) that has complementarity sequences to SNORD90. NRG3 was ranked as one of highest genes housing a complementarity sequence to SNORD90 without any mismatches. To increase our confidence in target selection we also utilized the C/D box snoRNA target prediction algorithm PLEXY, using default parameters (18). SNORD90 mature sequence was used as the input snoRNA and target sequence input was the entire human genome downloaded from the Ensembl genome browser database (Supplementary File 6-7). Prediction was conducted using whole transcriptomics (Supplementary File 6) as well as a targeted analysis for NRG3 (Supplementary File 7). Using the PLEXY algorithm also identified NRG3 as a possible target for SNORD90 (Supplementary File 6). Although NRG3 was not ranked among the highest predicted targets using the PLEXY algorithm, NRG3 was selected as the candidate gene target since it was the only gene that displayed overlap between the two target prediction methods (Supplementary File 5-6). PLEXY further reviled that SNORD90 has multiple possible interaction sites on NRG3, thus we selected the top three predicted sites for consideration in future experiments (figure 3A & Supplementary File 6-7).”

A major unanswered question in the manuscript is that, whereas SNORD90 binds to intronic sequences, methylation of mature NRG3 mRNA is also elevated by SNORD90 overexpression. Is methylation of NRG3 pre-mRNA restricted to the vicinity of SNORD90 binding, or are pre-mRNAs also methylated in exons? Addressing this question could provide greater clarity into the proposed mechanism of action of SNORD90 and validate its significance.

We hypothesized that the methylation of NRG3 pre-mRNA is not restricted to the SNORD90 binding site. We think that the secondary structure of pre-mRNA can facilitate depositing m6a to exonic regions. This is primarily supported by our observation of increased m6A abundance observed on both the pre-mRNA and mRNA of NRG3 (i.e. the increase of m6A seems to be preserved between pre-mRNA and mRNA). The literature has also shown that most m6a modifications are located in exons and is retained from pre-mRNA to mRNA [ke at el., 2017; PMID: 28637692].